# A high-resolution map of transcriptional repression

**Ziwei Liang**[1,2], **Karen E Brown**[1,2], **Thomas Carroll**[2], **Benjamin Taylor**[1,2†],
**Isabel Ferreirós Vidal**[1,2], **Brian Hendrich**[3,4], **David Rueda**[5,6], **Amanda G Fisher**[1,2],
**Matthias Merkenschlager**[1,2,6*]

[1]Lymphocyte Development Group, MRC London Institute of Medical Sciences, Faculty of Medicine, Imperial College London, London, United Kingdom; [2]Epigenetics Section, MRC London Institute of Medical Sciences, Faculty of Medicine, Imperial College London, London, United Kingdom; [3]Wellcome Trust – Medical Research Council Stem Cell Institute, Cambridge, United Kingdom; [4]Department of Biochemistry, University of Cambridge, Cambridge, United Kingdom; [5]Single Molecule Imaging Group, MRC London Institute of Medical Sciences, Faculty of Medicine, Imperial College London, London, United Kingdom; [6]Integrative Biology Section, MRC London Institute of Medical Sciences, Faculty of Medicine, Imperial College London, London, United Kingdom

**Abstract** Turning genes on and off is essential for development and homeostasis, yet little is known about the sequence and causal role of chromatin state changes during the repression of active genes. This is surprising, as defective gene silencing underlies developmental abnormalities and disease. Here we delineate the sequence and functional contribution of transcriptional repression mechanisms at high temporal resolution. Inducible entry of the NuRD-interacting transcriptional regulator Ikaros into mouse pre-B cell nuclei triggered immediate binding to target gene promoters. Rapid RNAP2 eviction, transcriptional shutdown, nucleosome invasion, and reduced transcriptional activator binding required chromatin remodeling by NuRD-associated Mi2beta/CHD4, but were independent of HDAC activity. Histone deacetylation occurred after transcriptional repression. Nevertheless, HDAC activity contributed to stable gene silencing. Hence, high resolution mapping of transcriptional repression reveals complex and interdependent mechanisms that underpin rapid transitions between transcriptional states, and elucidates the temporal order, functional role and mechanistic separation of NuRD-associated enzymatic activities.

**\*For correspondence:** matthias. merkenschlager@imperial.ac.uk

**Present address:** †AstraZeneca, Cambridge Science Park, Cambridge, United Kingdom

**Competing interests:** The authors declare that no competing interests exist.

## Introduction

Transcriptional gene activation and repression are essential for cell commitment, differentiation and homeostasis in metazoans. The dynamics of inducible gene expression have been characterised in exquisite detail (*Ballaré et al., 2013*; *Cosma, 2002*; *Grøntved et al., 2013*; *Vicent et al., 2011*; *Voss and Hager, 2014*; *Hargreaves et al., 2009*; *Ramirez-Carrozzi et al., 2009*; *Smale and Natoli, 2014*). Compared to gene activation, the mechanisms that underpin gene repression have received less attention (*Brockdorff and Turner, 2015*; *Cosma, 2002*; *Feng et al., 2016*; *Fuda et al., 2009*; *Grossniklaus and Paro, 2014*; *Su et al., 2004*; *Trinh et al., 2001*). Understanding transcriptional repression is important because defects in gene silencing result in developmental abnormalities and disease (*Denslow and Wade, 2007*; *Watson et al., 2012*). The nucleosome remodeling and deacetylase (NuRD) corepressor complex is critical for developmental gene regulation (*Costa et al., 2012*; *Laugesen and Helin, 2014*; *Reynolds et al., 2012*; *Yamada et al., 2014*). NuRD malfunction leads

to birth defects (*Fahrner and Bjornsson, 2014*) and underlies disease states ranging from autism spectrum disorders (*Yamada et al., 2014*; *Yang et al., 2016*) to cancer (*Laugesen and Helin, 2014*). NuRD is unique among corepressors in that it brings together histone deacetylation and ATP-dependent chromatin remodeling activities in the same complex, and therefore may have the necessary enzymatic attributes to convert active promoters into densely packed, hypoacetylated chromatin refractory to the transcriptional machinery. It has been argued that understanding the kinetics of changes in nucleosome positioning and modifications that accompany transcriptional repression is important to link the biochemical properties of NuRD with our understanding of chromatin-based gene regulation (*Denslow and Wade, 2007*). Meeting this challenge is difficult in experimental systems where the kinetics are slow (*Costa et al., 2012*), or gene silencing is asynchronous (*Reynolds et al., 2012*).

To investigate the sequence and causality of transcriptional repression mechanisms we focus on target genes of the NuRD-interacting transcriptional regulator Ikaros in B cell progenitors (*Ferreirós-Vidal et al., 2013*). Ikaros is essential for B cell development (*Heizmann et al., 2013*; *Joshi et al., 2014*; *Schwickert et al., 2014*) and frequently mutated in B cell malignancies (*Mulligan et al., 2008*, *2009*). *Myc* and the prototypic pre-B cell gene *Igll1* are direct Ikaros target genes (*Ferreirós-Vidal et al., 2013*; *Ma et al., 2010*; *Thompson et al., 2007*). *Igll1* encodes an essential component of the pre-B cell receptor (pre-BCR), which initially drives pre-B cell proliferation (*Melchers et al., 1993*), but subsequently triggers the expression of the Ikaros paralog Aiolos (*Thompson et al., 2007*). Together, Ikaros and Aiolos orchestrate the silencing of *Igll1*, the termination of pre-BCR expression, cell cycle exit, and differentiation of pre-B cells (*Sabbattini et al., 2001*; *Thompson et al., 2007*). *Igll1* therefore is part of a regulatory circuit that controls the transition from self-renewal to differentiation of B cell progenitors. *Myc* is highly expressed in proliferating B cell progenitors and can interfere with B cell progenitor differentiation by overriding cell cycle exit (*Ma et al., 2010*).

The interaction of Ikaros with the CHD4/Mi-2beta chromatin remodeling subunit of the NuRD corepressor complex (*Kim et al., 1999*) is direct (*Sridharan and Smale, 2007*) and evolutionarily conserved (*Kehle et al., 1998*). Although histone deacetylation has been linked to Ikaros-mediated repression (*Koipally et al., 1999*), the mechanisms of Ikaros-mediated gene regulation remain incompletely understood. At the *Igll1* promoter the transcriptional activator EBF1 and Ikaros are thought to compete with each other for binding to a composite Ikaros/EBF1 site (*Sabbattini et al., 2001*; *Thompson et al., 2007*). Overlap between binding sites for Ikaros and transcriptional activators is also found the Ikaros target gene *Dntt* (*Ernst et al., 1996*; *Hahm et al., 1998*; *Trinh et al., 2001*), and the Ikaros-related Hunchback protein competes for DNA binding with transcriptional activators at Hox genes in *Drosophila* (*Lawrence, 1992*). The concept of Ikaros-activator competition is supported by in vitro gel mobility shift experiments (*Thompson et al., 2007*; *Trinh et al., 2001*), but it remains unclear how such competition may play out in living cells.

Ikaros binds sequence motifs of gamma satellites, the core repeat unit of mouse pericentromeric heterochromatin (*Brown et al., 1999*, *1997*; *Cobb et al., 2000*; *Hahm et al., 1998*; *Klug et al., 1998*). Silent genes are often positioned within transcriptionally repressive compartments of the nucleus (*Bickmore, 2013*), including pericentromeric heterochromatin (*Brown et al., 1999*, *1997*), and the repositioning of *Rag* and *Dntt* loci accompanies stable, but not transient gene silencing (*Brown et al., 1999*; *Su et al., 2004*). These data suggested the attractive model that Ikaros may position silent genes close to pericentromeric heterochromatin (*Brown et al., 1999*; *Cobb et al., 2000*). Surprisingly, however, a direct role for Ikaros or NuRD in the repositioning of target genes has not been demonstrated (*Ferreirós-Vidal et al., 2013*; *Thompson et al., 2007*).

To study transcriptional repression at high temporal resolution we utilize the regulated translocation of Ikaros from the cytoplasm to the nucleus (*Ferreirós-Vidal et al., 2013*). Nuclear translocation results in immediate binding to target gene promoters, providing a clear path to target gene regulation. Using this approach we demonstrate that transcriptional repression is surprisingly rapid: promoters are depleted of RNAP2 and invaded by nucleosomes within minutes. Mechanistically, RNAP2 eviction is controlled by chromatin accessibility, not *vice versa*. Kinetic analysis reveals an unexpected temporal dissociation between NuRD enzymatic activities: CHD4-dependent remodeling - but not HDAC activity - initiates repression by excluding transcriptional activators and RNAP2. However, stable silencing and pericentromeric repositioning of Ikaros target genes require both nucleosome remodeling and HDAC activity. Hence, our high resolution view reveals that transcriptional

repression involves complex and interdependent mechanisms, and that the enzymatic activities of CHD4 and HDACs are selectively required at different times and for discrete aspects of gene silencing.

## Results

### In vivo dynamics of Ikaros and EBF1 binding at the Ikaros target gene promoter *Igll1*

Ikaros-ERt2 fusion proteins are retained in the cytoplasm of pre-B cells (*Ferreirós-Vidal et al., 2013*) until their release by the ERt2 ligand 4-hydroxytamoxifen (4-OHT), which initiates the translocation of Ikaros-ERt2 within minutes (*Figure 1A*, *Figure 1—figure supplement 1A*). Association with DAPI-dense pericentromeric gamma-satellite regions containing Ikaros binding sites (*Brown et al., 1999*, *1997*; *Hahm et al., 1998*) suggests that Ikaros binds DNA immediately after nuclear translocation. We exploited the temporal resolution afforded by inducible nuclear translocation to map the in vivo binding dynamics of Ikaros and EBF1 at the *Igll1* promoter, and to experimentally test whether Ikaros and EBF1 compete with each other (*Sabbattini et al., 2001*; *Thompson et al., 2007*).

ChIP and quantitative PCR showed a steep rise in Ikaros binding to *Igll1* promoter within 5 min (*Figure 1B*). The *Igll1* promoter has two Ikaros sites, Ik1 at −83nt and Ik2 at −107nt from the *Igll1* TSS. An EBF motif is embedded in the Ik1 site, and gel mobility shift experiments demonstrate mutually exclusive binding of EBF1 and Ikaros (*Thompson et al., 2007*). Despite the presence of two additional EBF1 motifs at −198 and −228nt, the embedded EBF1 site is critical for *Igll1* promoter activity (*Mårtensson and Mårtensson, 1997*) (*Figure 1C*). It was therefore surprising that the rise in Ikaros binding 5 min after 4-OHT was not accompanied by a corresponding fall in EBF1 binding (*Figure 1B*). Increased Ikaros binding without simultaneous eviction of EBF1 5 min after 4-OHT suggests that Ikaros does not directly displace EBF1.

We used kinetic modeling to explore the impact of increased nuclear Ikaros in the presence of a constant concentration of EBF1 and a constant number of *Igll1* promoter sites. In a 2-state model where DNA binding sites are either occupied by EBF1 or by Ikaros (EBF-DNA <-> Ikaros-DNA), any increase in Ikaros-DNA binding comes at the expense of EBF-DNA binding (*Figure 1D*). The observed temporal dissociation between Ikaros binding and EBF1 eviction by ChIP (*Figure 1B*) is inconsistent with direct competition between Ikaros (red) and EBF1 (green).

Interactions of transcription factors with target DNA are often transient, and not all target sites for DNA binding factors are occupied at any given time (*Elf et al., 2007*; *McNally et al., 2000*; *Poorey et al., 2013*; *Voss and Hager, 2014*). We therefore considered a 3-state model (*Figure 1E*) where in addition to sites occupied by EBF1 or Ikaros a fraction of DNA binding sites is free at any given time (EBF-DNA, green <-> free DNA, turquoise <-> Ikaros-DNA, red). In this scenario, the initial increase in Ikaros binding does not displace EBF1 directly, but rather reduces the fraction of unbound sites. Eventually, the reduced availability of unbound sites impacts on EBF1 by reducing the chances of successful re-binding during dynamic cycles of unbinding and re-binding. Consistent with this prediction, EBF1 ChIP measured a gradual reduction in EBF1 binding between 15 and 60 min after 4-OHT (*Figure 1B*). This revised model explains how increased Ikaros occupancy of the *Igll1* promoter does not affect EBF1 binding immediately, but with a significant delay relative to the observed increase in Ikaros binding.

The abundance of nuclear Ikaros rises gradually over time (*Figure 1A*, *Figure 1—figure supplement 1A*), yet ChIP shows that Ikaros binding flattens out after 5 min and shows little or no further increase after 15 min, even though declining EBF1 binding should release potential binding sites (*Figure 1B*). While the models in *Figure 1C and D* assume a one-step increase in nuclear Ikaros, modeling an incremental rise in nuclear Ikaros predicted a more pronounced increase in Ikaros binding (red) with little or no decrease in EBF1 binding (green) at equilibrium (*Figure 1—figure supplement 1B*). This does not reflect the experimental observations of a blunted rise in Ikaros binding and reduced EBF binding. The data were captured closely, however, when we allowed for the possibility that in response to the nuclear translocation of Ikaros the pool of DNA target sites available for binding decreases over time (*Figure 1—figure supplement 1C*, blue). We therefore investigated the impact of Ikaros on the chromatin state of target promoters.

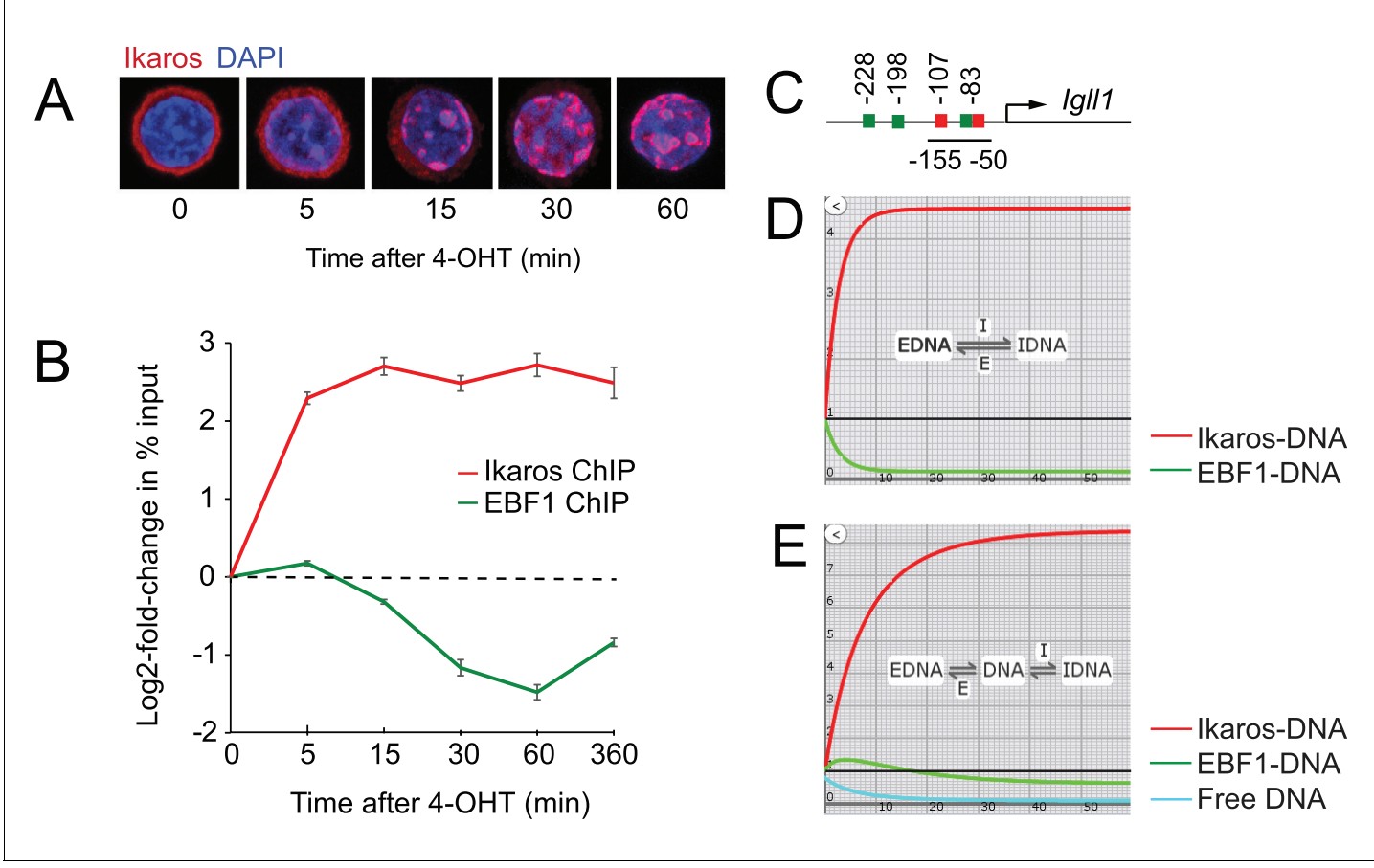

**Figure 1.** Ikaros binding and EBF eviction have different kinetics. (A) Time course of nuclear translocation of HA-Ikaros-ERt2 induced by 4-OHT. The fraction of nuclear Ikaros-ERt2 was estimated by immunofluorescence staining for the HA tag (red) as <5%, 20%, 50%, 80% and 95% at 0, 5, 15, 30 and 60 min after 4-OHT in 3 independent biological replicates. (B) ChIP-PCR time course of Ikaros and EBF1 binding at the *Igll1* promoter after 4-OHT. ChIP was done with antisera against the Ikaros C-terminus that detect Ikaros-ERt2 as well as pre-bound endogenous Ikaros. Log2 of mean ± SE, 3 independent biological replicates. Increased binding of Ikaros was significant by 5 min (p<0.05, Students' T test). Decreased binding of EBF1 was significant by 15 min. (C) The arrangement of binding sites for Ikaros (red) and EBF1 (green) at the *Igll1* promoter. (D) Two-state model with DNA binding sites occupied either by EBF1 or by Ikaros (EBF-DNA <=> Ikaros-DNA). Increase in Ikaros binding (red) comes at the expense of EBF binding (green). (E) Three-state model where a fraction of DNA binding sites is unbound at any moment in time (turquoise).

The following source data and figure supplement are available for figure 1:

**Source data 1.** Numerical data used to generate *Figure 1B*.

**Figure supplement 1.** Nuclear translocation of Ikaros and 3- versus 4-state models of Ikaros-EBF competition.

## Nuclear translocation of Ikaros evicts RNAP2 and positions nucleosomes that abrogate *Igll1* promoter accessibility

ChIP sequencing (ChIP-seq) showed that the nuclear translocation of Ikaros abolished RNAP2 binding to promoters and gene bodies of the target genes *Igll1* and *Vpreb1* (*Figure 2A*, *Figure 2—figure supplement 1*) that are repressed by Ikaros (*Ferreirós-Vidal et al., 2013*; *Thompson et al., 2007*). RNAP2 occupancy remained unchanged at the nearby *Top3b* promoter, which is not regulated by Ikaros (*Figure 2A*, *Figure 2—figure supplement 1*).

We mapped the nucleosome landscape before and after nuclear translocation of Ikaros using micrococcal nuclease digestion and sequencing (MNase-seq). Prior to nuclear translocation of Ikaros the *Igll1* promoter showed a nucleosome-free region that was occupied by RNAP2. Upon nuclear translocation of Ikaros this region became occluded by nucleosomes, and RNAP2 occupancy was

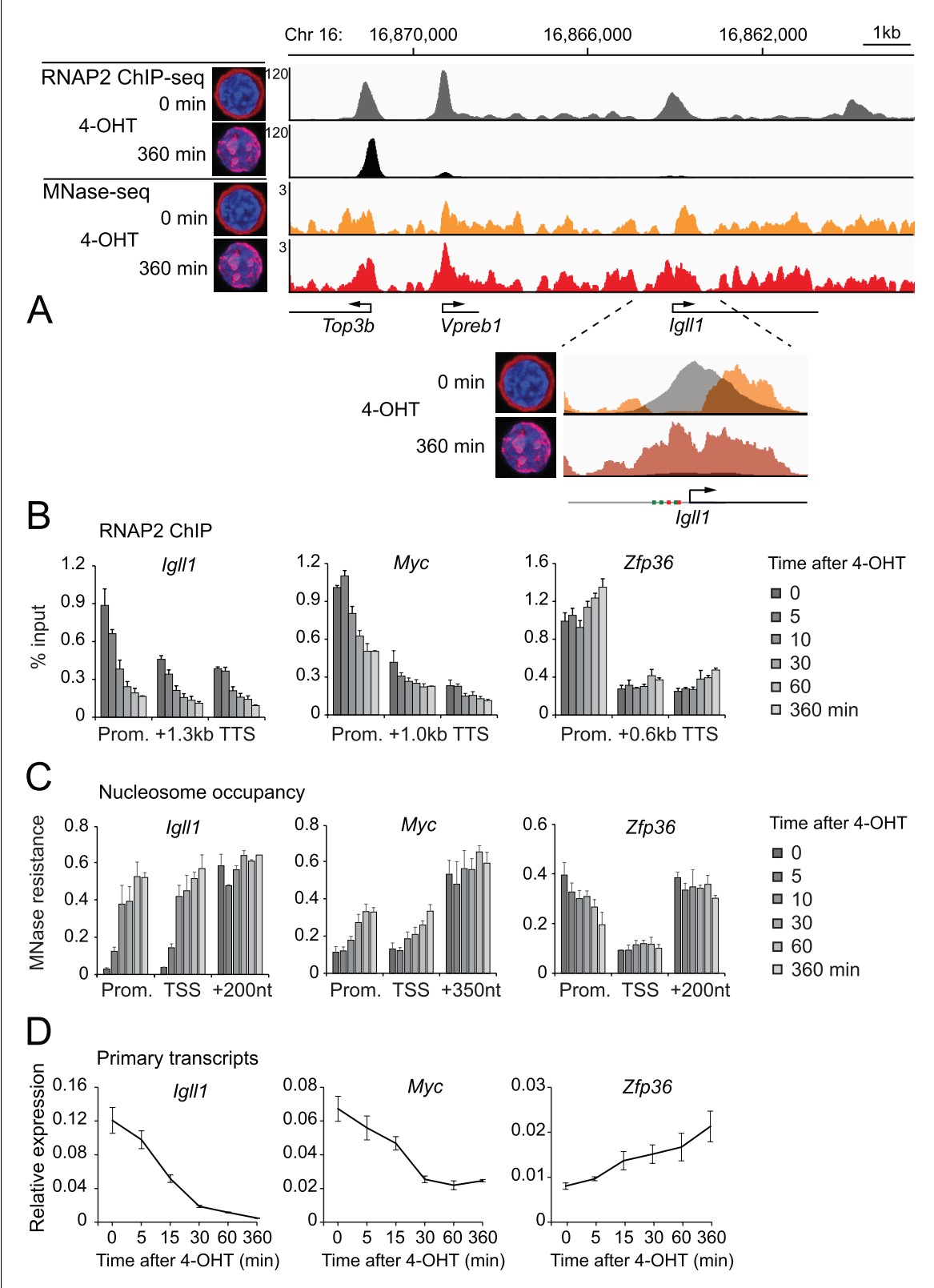

**Figure 2.** Kinetics of Ikaros-mediated changes in RNAP2 occupancy, locus accessibility and transcription. (**A**) Ikaros triggers RNAP2 eviction and nucleosome recruitment at the *Igll1* promoter. RNAP2 ChIP-seq: pooled data from 4 independent biological replicates. MNase-seq: pooled data from 3 independent biological replicates. *Figure 2—figure supplement 1* shows the data for each individual replicate. (**B**) Kinetics of RNAP2 binding at *Igll1*, *Myc*, and *Zfp36* after 4-OHT by ChIP-PCR. Mean ± SE, n = 3. RNAP2 binding was significantly reduced at the *Igll1* promoter, gene body and TTS

*Figure 2 continued on next page*

*Figure 2 continued*

from 15 min and the *Myc* promoter from 30 min. (C) Kinetics of nucleosome occupancy at *Igll1, Myc*, and *Zfp36* after 4-OHT by MNase-PCR of 80–120 bp amplicons. 3 independent biological replicates. Nucleosome occupancy was significantly increased at the *Igll1* promoter and TSS from 5 min and the *Myc* promoter and TSS from 15 min. (D) Kinetics of primary *Igll1, Myc*, and *Zfp36* transcripts after 4-OHT by RT-PCR with primer pairs designed to span an intron-exon boundary (*Igll1* and *Myc*) or located within an intron (*Zfp36*). Mean ± SE, 4 independent biological replicates. Primary *Igll1* and *Myc* transcripts were significantly reduced from 15 and 5 min, respectively.

The following source data and figure supplements are available for figure 2:

**Source data 1.** Numerical data used to generate *Figure 2B,C and D*.
**Figure supplement 1.** Reproducibility of RNAP2 ChIP-seq and MNase-seq.
**Figure supplement 1—source data 1.** Numerical data used to generate *Figure 2—figure supplement 1B*.

reduced (*Figure 2A*, inset and *Figure 2—figure supplement 1*). Reduced accessibility of the *Igll1* promoter is consistent with models where Ikaros and EBF1 compete for a decreasing number of accessible binding sites (*Figure 1—figure supplement 1C*).

## Loss of RNAP2, reduced promoter accessibility, and transcriptional repression are early and near-simultaneous events

We next explored the dynamics of Ikaros-induced changes in RNAP2 occupancy, chromatin organisation and transcription at the Ikaros-repressed genes *Igll1* and *Myc*. ChIP-PCR demonstrated a rapid decrease in RNAP2 occupancy at the promoters, gene bodies and transcription termination sites (TTS) of Ikaros-repressed genes (*Figure 2B*). RNAP2 occupancy decreased at promoters, gene bodies, and the 3' end of transcription units (*Figure 2B*), suggesting that Ikaros represses *Igll1* and *Myc* transcription by interfering with RNAP2 recruitment rather than elongation. Genome-wide, RNAP2 occupancy was significantly reduced at 372 downregulated genes 6 hr after Ikaros

**Table 1.** Ikaros represses transcription by interfering with RNAP2 recruitment rather than elongation. Analysis of differential gene expression (adj. p<0.05) 360 min after 4-OHT-induced nuclear translocation of Ikaros in B3 cells, RNAP2 ChIP-seq, and the distribution of RNAP2 over the TSS and the gene body derived from RNAP2 ChIP-seq. RNAP2 occupancy was significantly decreased at only 40.3% (372 of 924) of genes downregulated at adj. p<0.05 but increased to 62% when considering only genes downregulated with adj. p<0.01 and a minimal log2 fold-change of 2. This suggests that the failure to detect decreased RNAP2 occupancy at the majority of downregulated genes may be due to the limited sensitivity of RNAP2 ChIP-seq compared to RNA-seq.

| Gene expression | RNAP2 occupancy by ChIP | | | |
|---|---|---|---|---|
| | Global | TSS~gene body | TSS>gene body | TSS<gene body |
| Unchanged: 8799 | Unchanged: 8125 | 8019 | 49 | 57 |
| | Reduced: 405 | 378 | 11 | 16 |
| | Increased: 269 | 256 | 10 | 3 |
| Downregulated by Ikaros: 924 | Unchanged: 549 | 520 | 12 | 17 |
| | Reduced: 372 | 324 | 15 | 33 |
| | Increased: 3 | 3 | 0 | 0 |
| Upregulated by Ikaros: 855 | Unchanged: 634 | 615 | 15 | 4 |
| | Reduced: 6 | 6 | 0 | 0 |
| | Increased: 215 | 200 | 13 | 2 |

**Source data 1.** Numerical data used to generate *Table 1*.

translocation (*Table 1*). The great majority of repressed genes with reduced RNAP2 occupancy showed a global reduction in RNAP2 binding to both the promoter and the gene body (324 of 372 or 87%). Only a small minority of repressed genes showed a block in RNAP2 elongation, as judged by preferential reduction of RNAP2 occupancy over the gene body (15 of 372 or 4% of genes with reduced RNAP2 occupancy). Hence, transcriptional repression of Ikaros target genes occurred by reduced RNAP2 recruitment, and a block in RNAP2 elongation was the exception.

We assessed nucleosome occupancy of promoters and transcription start sites by PCR amplification of fragments protected from MNase digestion. Nucleosome occupancy increased with similar kinetics as RNAP2 eviction (*Figure 2C*).

We used quantitative RT-PCR with primers spanning intron-exon boundaries to quantify the abundance of unspliced (primary) transcripts, which are short-lived and serve as an indicator of transcriptional activity (*Figure 2D*). The abundance of *Igll1* and *Myc* primary transcripts was reduced within minutes of Ikaros translocation with similar kinetics as RNAP2 eviction and increased nucleosome occupancy (*Figure 2D*). These changes were selective for Ikaros-repressed genes, as illustrated by comparison with the Ikaros-induced gene *Zfp36* (*Figure 2C–D*).

## RNAP2 is not required for maintaining chromatin accessibility at Ikaros target gene promoters

We next asked whether nucleosome invasion of Ikaros-repressed target promoters was secondary to reduced RNAP2 occupancy (*Gilchrist et al., 2012*) (*Figure 3A*, right). Triptolide triggers the global degradation of RNAP2 (*Manzo et al., 2012*; *Wang et al., 2011*) and resulted in the near-complete loss of RNAP2 from whole cell lysates (*Figure 3A*, left) and from the *Igll1* and *Myc* promoters (*Figure 3B*, left; the inactive *Rex1* promoter is shown as a negative control). Loss of RNAP2 reduced promoter occupancy by the basal transcription factor TFIIB (*Figure 3B*, center) but, interestingly, the *Igll1* and *Myc* promoters remained sensitive to MNase in the absence of RNAP2 (*Figure 3B*, right). Hence, accessibility of Ikaros target gene promoters is not simply a consequence of RNAP2 occupancy.

We next compared the impact of Ikaros on target gene promoters in control and RNAP2-depleted cells. Nuclear translocation of Ikaros-ERt2 efficiently reduced the accessibility of the *Igll1* and *Myc* promoters even after RNAP2 depletion (*Figure 3C*).

To address the formal possibility that the ERt2 component of the Ikaros-ERt2 fusion protein contributed to the Ikaros-induced reduction in promoter accessibility we used a complementary approach where the nuclear translocation of Ikaros is triggered by the inducible cleavage of Ikaros-TEV-ERt2 fusion proteins (*Wehr et al., 2006*) (*Figure 3—figure supplement 1A*). This confirmed that nuclear translocation of Ikaros in the absence of ERt2 was able to regulate promoter accessibility (*Figure 3—figure supplement 1B*).

The demonstration that the *Igll1* and *Myc* promoter regions remain accessible in the absence of RNAP2 mechanistically separates RNAP2 occupancy from promoter accessibility. We therefore focused on the question whether Ikaros controls promoter accessibility through active chromatin remodelling.

## Ikaros controls promoter accessibility through the NuRD-associated chromatin remodeller Mi-2β/CHD4

We used RNA interference to deplete the NuRD-associated ATPase subunit Mi-2β/CHD4 encoded by *Chd4* (*Figure 4A*). CHD4 depletion substantially reduced the impact of Ikaros on the accessibility of the *Igll1* and *Myc* promoters (*Figure 4B*), indicating a key role for CHD4 in mediating Ikaros-induced nucleosome remodeling at the *Igll1* and *Myc* promoters. Ikaros-induced RNAP2 eviction and transcriptional repression were significantly delayed by *Chd4*-RNAi (*Figure 4C*), consistent with a role for CHD4-dependent chromatin remodeling in the silencing of Ikaros target genes.

We next assessed the impact of Ikaros on the association of NuRD components with the *Igll1* promoter. ChIP-PCR showed a rapid increase in the binding of the NuRD subunits CHD4 and MBD3 (*Figure 4D*). The onset CHD4 and MBD3 recruitment mirrored increased Ikaros binding after 4-OHT (*Figure 1B*). In contrast to Ikaros, increased binding of CHD4 and MBD3 was transient, returning close to their starting level by 6 hr (*Figure 4D*).

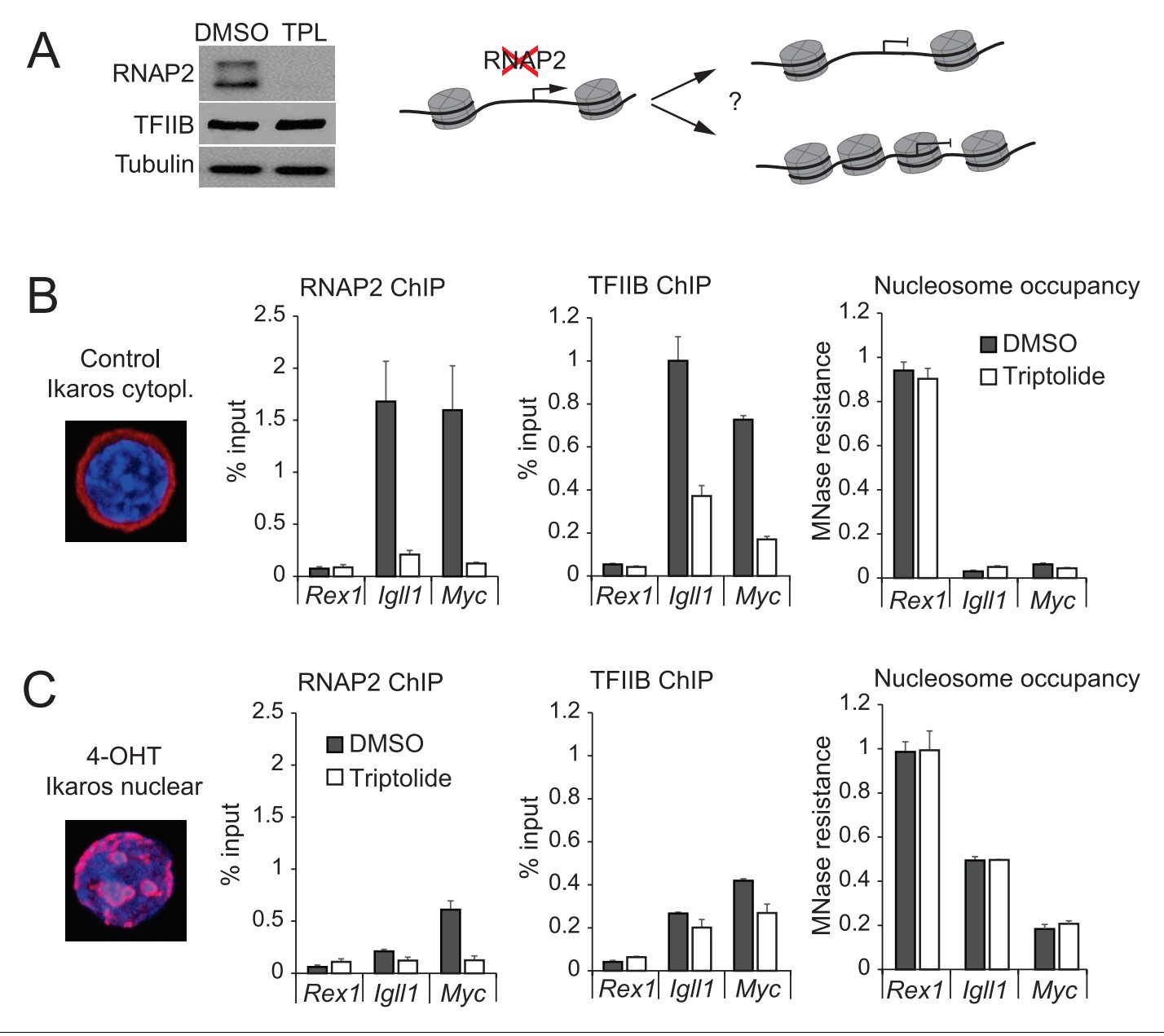

**Figure 3.** RNAP2 is not required for chromatin accessibility of Ikaros target promoters. (A) Left: Western blotting for RNAP2 and TFIIB after 3 hr treatment with 1 uM triptolide (TPL). Tubulin is a loading control. 3 independent biological replicates. Right: Possible impact of RNAPII removal: promoters may remain nucleosome-free (top) or not (bottom). (B) RNAP2 ChIP-PCR, TFIIB ChIP-PCR and MNase-PCR at the indicated promoters following 4 hr of 1 μM triptolide or carrier (DMSO, grey). Mean ± SE, 3 independent biological replicates. Triptolide significantly reduced RNAP2 and TFIIB occupancy but did not significantly affect nucleosome occupancy at the *Igll1* and *Myc* promoters. (C) As (B) with 4-OHT present during the last hour. Mean ± SE, 3 independent biological replicates. Triptolide did not significantly affect the ability of Ikaros to increase nucleosome occupancy at the *Igll1* and *Myc* promoters.

The following source data and figure supplements are available for figure 3:

**Source data 1.** Numerical data used to generate *Figure 3B and C*.

**Figure supplement 1.** Nuclear translocation of Ikaros by proteolytic cleavage of Ikaros fusion proteins.

**Figure supplement 1—source data 1.** Numerical data used to generate *Figure 3—figure supplement 1A,B,C and D*.

## The SWI/SNF-associated chromatin remodeler BRG1 is antagonistic with Ikaros, and promotes the activity of Ikaros-repressed genes

CHD4 and the SWI-SNF-associated chromatin remodeler BRG1 can have antagonistic effects on transcriptional regulation (*Curtis and Griffin, 2012*; *Gao et al., 2009*; *Ramirez-Carrozzi et al., 2006*). To assess the relationship between Ikaros and BRG1 we compared genes regulated by Ikaros (See *Table 1—source data 1*) with genes regulated by BRG1 (*Bossen et al., 2015*). Genes activated by BRG1 were repressed by Ikaros (Odds ratio = 2.97, p=$1.7 \times 10^{-30}$) and vice versa (Odds ratio = 3.32, p=$9.5 \times 10^{-31}$). Conversely, Ikaros-repressed genes were depleted among BRG1-repressed genes (Odds ratio = 0.62) and vice versa (Odds ratio = 0.31; *Table 2*). This analysis indicates antagonistic regulation of target genes by Ikaros and BRG1.

## NuRD recruitment and BRG1 eviction from Ikaros target gene promoters

At the peak of CHD4 and MBD3 binding, the *Igll1* promoter showed a marked decline in BRG1 binding (*Figure 4D*). BRG1 association with the *Igll1* promoter did not recover at later time points, when CHD4 and MBD3 binding returned to their starting levels (*Figure 4D*).

## Histone deacetylase activity is dispensable for Ikaros-induced loss of RNAP2, reduced promoter accessibility and transcriptional repression

In addition to the chromatin remodeler CHD4, the NuRD complex contains the histone deacetylases HDAC1/2 (*Dege and Hagman, 2014*). Given the critical role for NuRD-associated chromatin remodeling activity in Ikaros-mediated gene silencing we asked whether HDAC activity was equally involved. Unexpectedly, broad inhibition of HDAC activity by trichostatin A (TSA) did not detectably interfere with Ikaros-induced loss of RNAP2, reduced promoter accessibility and early transcriptional repression of *Igll1* and *Myc* (*Figure 5A*, *Figure 5—figure supplement 1A*). Application of the HDAC1/2 inhibitor MS-275 confirmed that Ikaros initiated the transcriptional downregulation of *Igll1* and *Myc* regardless of HDAC1/2 activity (*Figure 5B*, *Figure 5—figure supplement 1B*). Similarly, HDAC inhibition did not interfere with the transcriptional downregulation of Ikaros target genes in primary pre-B cells (*Figure 5C*). To address the possibility that the ERt2 component of the Ikaros-ERt2 fusion protein might account for the apparent lack of HDAC activity in Ikaros-initiated gene silencing we performed experiments where the nuclear translocation of Ikaros was triggered by the inducible cleavage of Ikaros-TEV-ERt2 fusion proteins. TEV-induced nuclear translocation of Ikaros in the absence or ERt2 confirmed that HDAC activity was not required for the loss of accessibility (*Figure 5D*, left) or the transcriptional silencing of *Igll1* and *Myc* (*Figure 5D*, right).

These data demonstrate that - unexpectedly - HDAC activity is dispensable for Ikaros-induced repression. Consistent with this conclusion, ChIP experiments showed that although Ikaros reduced the acetylation of histone H3 and H4 at the target genes *Igll1* and *Myc*, the kinetics of histone deacetylation were markedly slower than those observed for reduced promoter accessibility, RNAP2 eviction, and the decline in primary transcripts (*Figure 5—figure supplement 1C*). We saw no detectable increase in the trimethylation of H3K27 by PRC2 (*Oravecz et al., 2015*) at the *Igll1* and *Myc* promoters 24 hr after nuclear translocation of Ikaros (Data not shown). Histone modifications are therefore not likely drivers for early events in Ikaros-mediated repression.

## HDAC activity contributes to lasting repression and repositioning of Ikaros target genes to repressive nuclear compartments

While the immediate transcriptional downregulation of Ikaros target genes was unaffected by HDAC inhibition (*Figure 5*), we found that Ikaros target gene transcription was not stably silenced 24 hr after Ikaros nuclear translocation in the presence of TSA (*Figure 6A*).

We examined the protection of nucleosome-associated DNA at the *Igll1* promoter and TSS using PCR amplicons of defined length (*Figure 6B*, top). TSA treatment for 24 hr did not affect Ikaros-induced nucleosome protection of short amplicons comprising 90–110 base pairs centred on peaks of MNase (*Figure 6B*, bottom left). In contrast, TSA reduced Ikaros-induced protection of longer

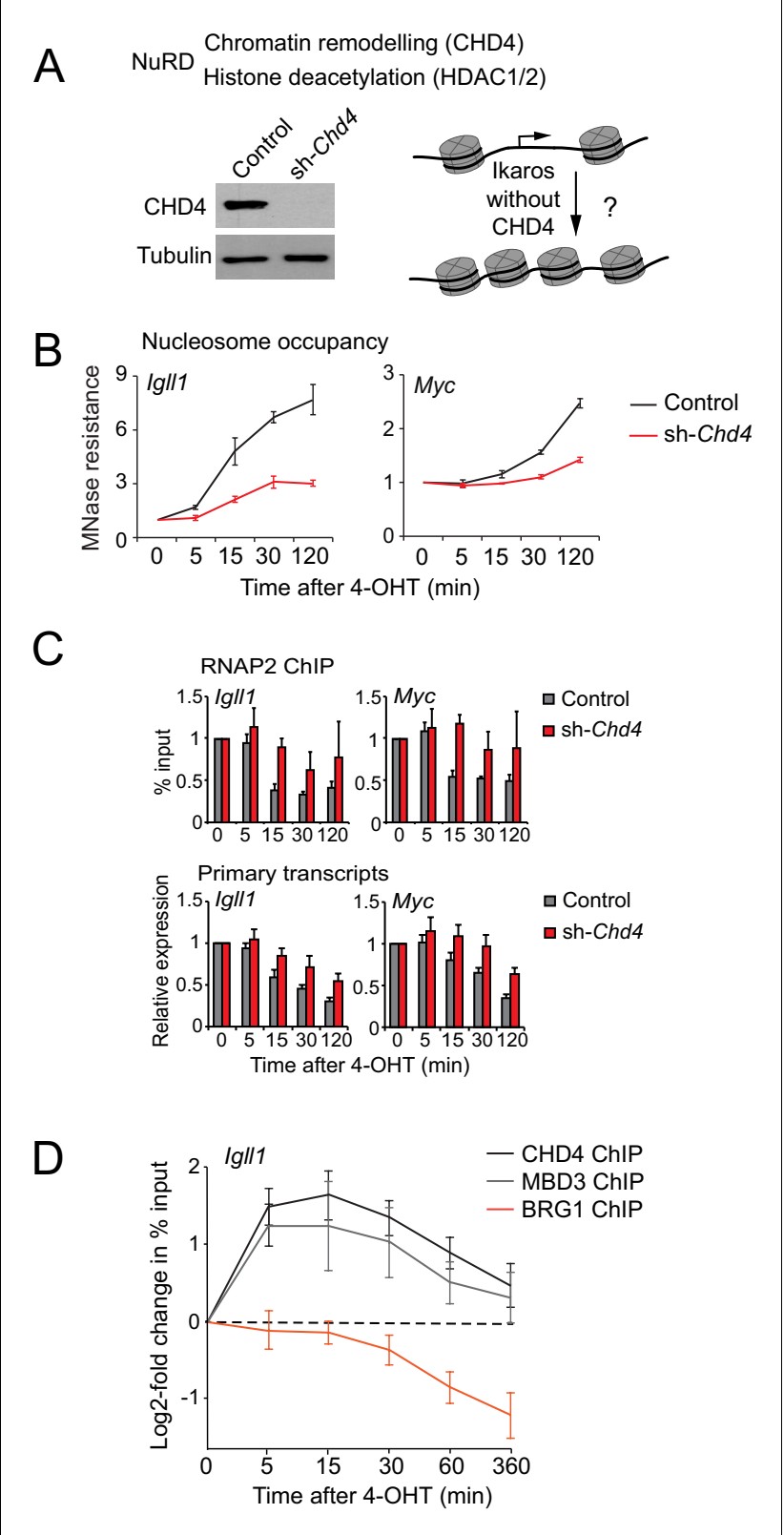

**Figure 4.** Ikaros controls promoter accessibility through NuRD-associated chromatin remodeling. (**A**) Left: CHD4 expression in control and *Chd4* shRNA cells by western blotting. Tubulin is a loading control. One of 5 independent biological replicates. Right: Experimental outline. (**B**) MNase-PCR at the *Igll1* and *Myc* promoters in control (black) or *Chd4* shRNA cells (red) at the indicated times after 4-OHT. Mean ± SE, 3 independent biological

*Figure 4 continued*

replicates. *Chd4* shRNA significantly reduced the Ikaros-induced increase in nucleosome occupancy at 15, 30 and 120 min at the *Igll1* promoter and at 30 and 120 min at the *Myc* promoter. (**C**) RNAP2 ChIP-PCR (top) and MNase-PCR (bottom) at the *Igll1* and *Myc* promoters after 4-OHT in control (black) or Chd4 shRNA cells (red). Mean ± SE, 3 independent biological replicates. RNAP2 binding was significantly reduced in control cells but not in *Chd4* shRNA-treated cells from 5 to 120 min after 4-OHT at the *Igll1* and the *Myc* promoter. Primary transcripts were significantly reduced in control but not in *Chd4* shRNA-treated cells at 15 and 30 min for *Igll1* and at 30 and 120 min for *Myc*. (**D**) ChIP-PCR for CHD4 (black), MBD3 (grey) and BRG1 (orange) at the *Igll1* promoters at the indicated times after 4-OHT. Mean ± SE, 5 independent biological replicates for CHD4 and BRG1, 3 independent biological replicates for MBD3. CHD4 and MBD3 binding at the *Igll1* promoter were significantly increased from 5 to 60 min. BRG1 binding was significantly decreased from 30 to 120 min.

The following source data is available for figure 4:

**Source data 1.** Numerical data used to generate *Figure 4B,C and D*.

amplicons of 130–140 bp, which represent the near complete length of 147 DNA base pairs associated with full nucleosomes (*Figure 6B*, bottom right). These data suggest that in the absence of HDAC activity, nucleosomes at the *Igll1* promoter and TSS may remain unstable (*Henikoff et al., 2011*) or less evenly spaced. To address if this is due to missing H2A-H2B dimers we performed ChIP for histone H2B and H3. At the *Igll1* promoter and TSS, Ikaros triggered an increase in the occupancy not only of total histone H3, but also in the ratio of histone H2B over H3 (*Figure 6C*). Prolonged HDAC inhibition significantly blunted this increase in the H2B/H3 ratio (*Figure 6C*), indicating that HDAC activity facilitated the assembly of H2B-containing nucleosomes.

To test the role of Ikaros in the repositioning of target genes to the transcriptionally repressive environment of pericentromeric heterochromatin we carried out 3D DNA-FISH experiments. In the absence of 4-OHT, only a small minority (8%) of *Igll1* alleles (green) were associated with γ-satellite DNA (red), the repeat unit of mouse pericentromeric heterochromatin (*Figure 6D*). In contrast, most (78%) *Igll1* alleles were associated with γ-satellite DNA after 4-OHT (*Figure 6D*), indicating that increased nuclear Ikaros cells was sufficient for the repositioning of *Igll1* to pericentromeric heterochromatin. Ikaros-induced of *Igll1* repositioning was reduced to 55% of alleles by low doses (1 ng/ml) of TSA (*Figure 6D*). We note that under these conditions TSA effectively inhibited histone deacetylation (*Figure 5—figure supplement 1A*) but did not globally disrupt nuclear organisation, since centromeric heterochromatin retained its compacted state and association with the nuclear periphery (*Figure 6D*).

Taken together, incomplete transcriptional silencing of *Igll1*, partial MNase protection of *Igll1* promoter DNA and incomplete repositioning of *Igll1* alleles to pericentromeric heterochromatin in the presence of HDAC inhibitors indicate that HDAC activity is required for full *Igll1* silencing, even though it is dispensable for the early events in transcriptional repression (*Figure 5*).

## Interdependence of silencing mechanisms leveraged by Ikaros

The experiments discussed above suggest that Ikaros may utilize silencing mechanisms not just sequentially, but also in a mutually interdependent fashion. We therefore asked whether CHD4-

**Table 2.** BRG1 promotes the expression of Ikaros-repressed genes. Shown is the overlap in differential gene expression (adj. p<0.05) after 4-OHT-induced nuclear translocation of Ikaros in B3 cells and *Smarca4* deletion in B cell progenitors (***Bossen et al., 2015***).

|  | Activated by BRG1 | Repressed by BRG1 |
|---|---|---|
| Downregulated by Ikaros | Odds ratio = 2.97<br>p=$1.7 \times 10^{-30}$ | Odds ratio = 0.62<br>p=0.99 |
| Upregulated by Ikaros | Odds ratio = 0.31<br>p=1.00 | Odds ratio = 3.32<br>p=$9.5 \times 10^{-31}$ |

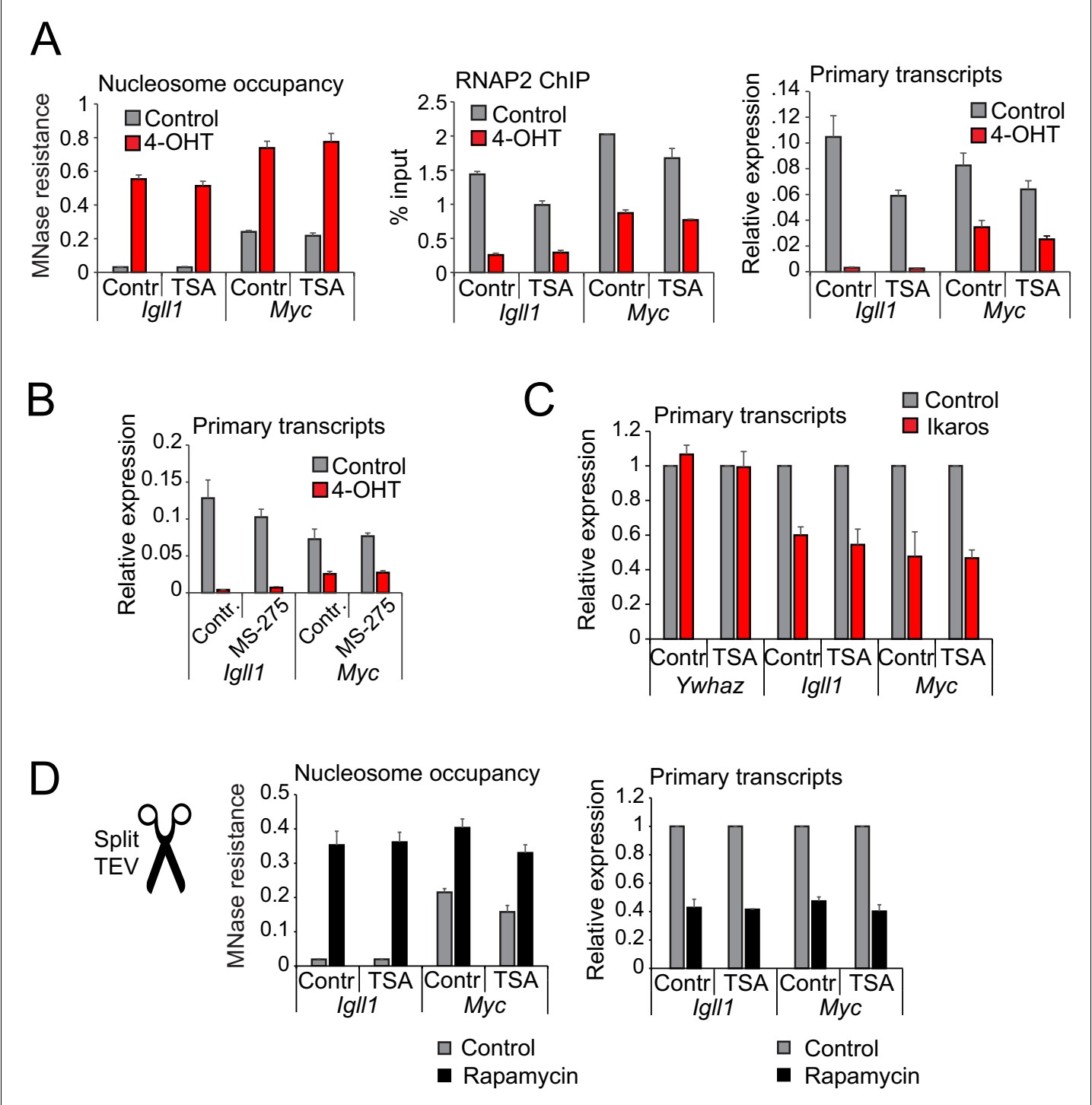

**Figure 5.** Histone deacetylase activity is dispensable for initiating transcriptional repression. (**A**) MNase-PCR (left), RNAP2 ChIP-PCR (middle) and RT-PCR (right). Where indicated B3 cells were treated for 1 hr with 1 ng/ml TSA and/or 4-OHT. *Figure 5—figure supplement 1A* shows TSA effects on histone acetylation. Mean ± SE, 3 independent biological replicates. TSA did not significantly affect Ikaros-induced changes in nucleosome occupancy and RNAP2 binding to the *Igll1* and *Myc* promoter, or *Igll1* primary transcripts. (**B**) RT-PCR for primary *Igll1* and *Myc* transcripts in control or MS-275-treated cells (1 hr, 10 uM) before (grey) and 1 hr after 4-OHT (red). Mean ± SE, 3 independent biological replicates. *Figure 5—figure supplement 1B* shows effects of MS-275 on histone acetylation. MS-275 treatment weakly affected Ikaros-induced changes in *Igll1* primary transcripts and did not significantly affect Ikaros-induced changes in *Myc* primary transcripts. (**C**) RT-PCR for *Igll1*, *Myc* and the housekeeping gene *Ywhaz* in primary pre-B cells transduced with control (IRES-GFP) or Ikaros (Ikaros-IRES-GFP). Where indicated cells were treated with 1 ng/ml TSA. Mean ± SE, 3 independent biological replicates. *Figure 5—figure supplement 1D* shows TSA effects on histone acetylation. TSA did not significantly affect Ikaros-induced

*Figure 5 continued on next page*

*Figure 5 continued*

changes in *Igll1* and *Myc* primary transcripts. (D) MNase-PCR (left) and RT-PCR (right) at *Igll1* and *Myc* in control or TSA-treated cells (2 hr, 1 ng/ml) before (grey) and 2 hr after TEV cleavage-induced nuclear translocation of Ikaros (black). Mean ± SE, 3 independent biological replicates.

The following source data and figure supplement are available for figure 5:

**Source data 1.** Numerical data used to generate *Figure 5A,B,C and D*.
**Figure supplement 1.** Changes in global and local histone acetylation in response to HDAC inhibitors and Ikaros.

mediated chromatin remodeling contributes to the repositioning of *Igll1* loci. We found that the Ikaros-induced repositioning of *Igll1* alleles was severely impaired (from 78% to 13%) by knockdown of *Chd4* (*Figure 7A*). Pericentromeric repositioning of Ikaros target loci therefore requires NuRD-associated chromatin remodeling as well as HDAC activity.

To explore the relationship between silencing mechanisms further, we revisited the models for competition between Ikaros and EBF1 for binding to the *Igll1* promoter (*Figure 1C,D*; *Figure 1—figure supplement 1C*). We compared the kinetics of Ikaros and EBF1 binding to *Igll1* in control cells (*Figure 7B* left) and in *Chd4*-depleted cells (*Figure 7B* right, sh*Chd4*). *Chd4*-depleted cells showed a steeper rise in Ikaros binding between 5 and 15 min than control cells. Strikingly, the decrease in EBF1 binding seen in control cells was significantly blunted in the absence of CHD4. In this regard, the binding kinetics of Ikaros and EBF1 in the absence of CHD4 resembled a 3-state model (*Figure 1—figure supplement 1B*) where an incremental increase in nuclear Ikaros concentration triggers a steep rise in Ikaros binding during the second time interval and results in little or no eviction of EBF1. In control cells, the observed dynamics more closely resembled a 4-state model (*Figure 1—figure supplement 1C*) where Ikaros and EBF1 compete for binding sites that remain accessible. Consistent with a scenario where Ikaros not only displaces EBF1, but ultimately limits its own binding, integration of MNase-seq with Ikaros ChIP-seq showed that nuclear translocation of Ikaros increases nucleosome occupancy at Ikaros ChIP-seq peaks genome-wide (*Figure 7C*).

## Discussion

Here we take a high-resolution view at transcriptional repression based on the inducible nuclear translocation of Ikaros. Nuclear translocation of Ikaros results in immediate binding to target gene promoters, providing a clear path to target gene regulation. Kinetic ChIP shows half-maximal binding of Ikaros to the *Igll1* target gene promoter within 5 min (*Figure 7D*). Unexpectedly, increased Ikaros binding did not directly displace the transcriptional activator EBF1 from the *Igll1* promoter, ruling out simple 2-state models where activator and repressor directly compete with each other. Ikaros effected half-maximal removal of RNAP2, transcriptional silencing and promoter invasion by nucleosomes within 12 min of nuclear translocation (*Figure 7D*).

Ikaros recruited the NuRD-associated chromatin remodeller CHD4 to target promoters, and CHD4 was required for the invasion of the *Igll1* and *Myc* promoters by nucleosomes, the exclusion of RNAP2, and the eviction of EBF1. Nuclear translocation of Ikaros reduced the accessibility of Ikaros binding sites genome-wide. Ultimately, Ikaros-induced chromatin remodeling limited the binding of the repressor itself, as evidenced by increased Ikaros binding after knockdown of *Chd4*. The binding kinetics of Ikaros and EBF1 at the *Igll1* promoter are best explained if a fraction of binding sites become inaccessible as a result of Ikaros nuclear translocation. Mechanistically, Ikaros can reduce EBF1 binding to the *Igll1* promoter only in the presence of CHD4, and EBF1 is binding is sensitive to local chromatin structure (*Treiber et al., 2010*).

The synchronous nature of our system enabled the detection of a transient increase in the association of the NuRD components CHD4 and MBD3 to Ikaros target promoters. Despite genetic evidence that NuRD is required for the differentiation of mouse embryonic stem cells, increased NuRD recruitment was not seen during this less synchronous process (*Reynolds et al., 2012*). Coincident with increased NuRD recruitment, the SWI/SNF-associated chromatin remodeler BRG1 was evicted from the *Igll1* promoter. Ikaros and BRG1 were broadly antagonistic in B cell progenitors, since BRG1 was required for the activity of many Ikaros-repressed genes. BRG1 did not return to the

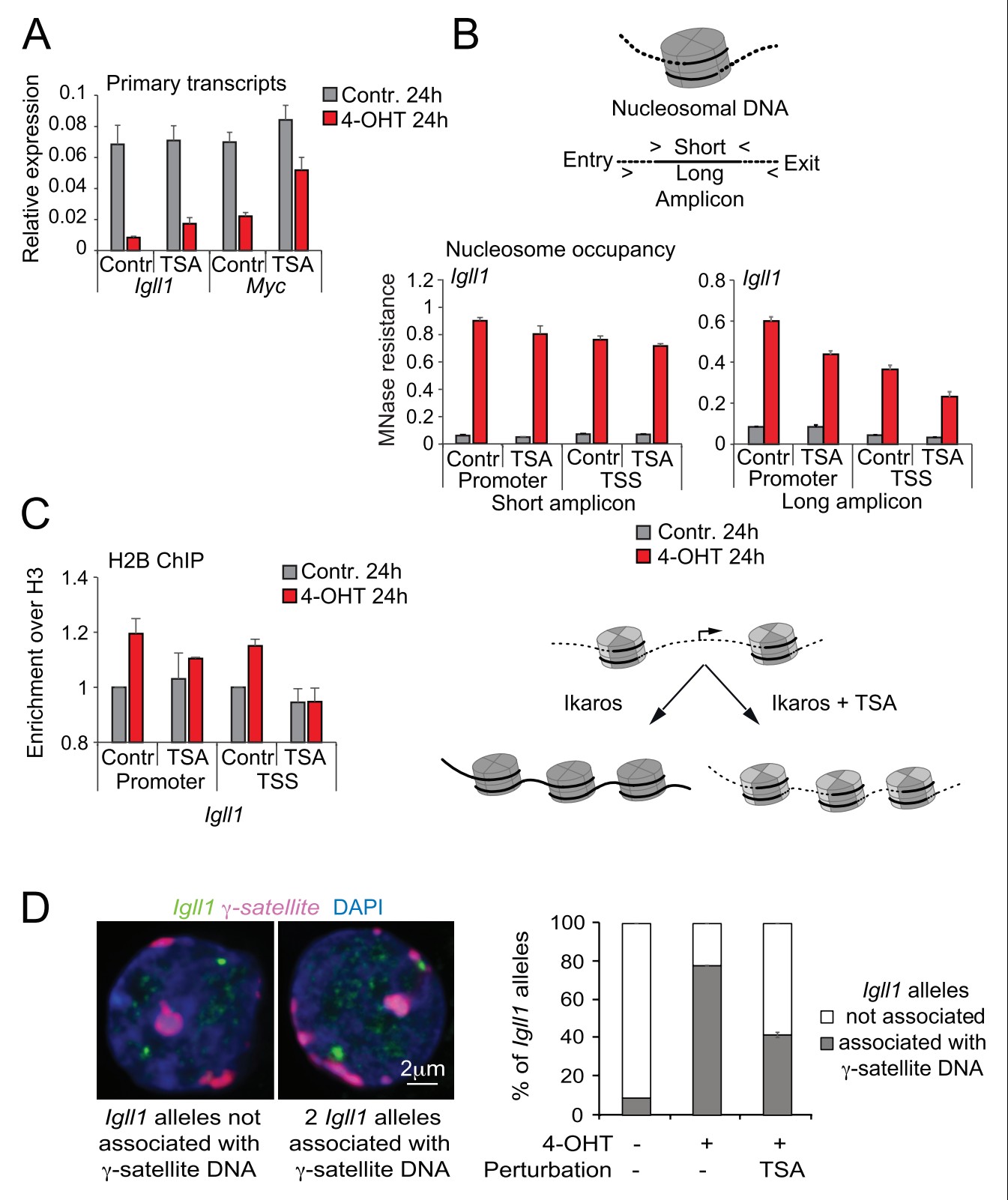

**Figure 6.** Histone deacetylation contributes to stable gene silencing. (**A**) RT-PCR showed that 1 ng/ml TSA for 24 hr significantly relieved Ikaros-induced reduced repression of *Igll* and *Myc* primary transcripts. Mean ± SE, 3 independent biological replicates. (**B**) MNase PCR showed that 1 ng/ml TSA for 24 hr did not significantly affect protection of 80–120 bp amplicons (short, left) but significantly reduced protection of 130–140 bp amplicons (long, right) at the *Igll1* promoter. Mean ± SE, 3 independent biological replicates. (**C**) ChIP-PCR to assess Ikaros-induced recruitment of histone H2B to the *Igll1*

*Figure 6 continued on next page*

*Figure 6 continued*
promoter between control cells and cells treated with 1 ng/ml TSA for 24 hr. Enrichment was normalised to total H3. Mean ± SE, 3 independent biological replicates. TSA significantly blunted the Ikaros-induced increase the H2B/H3 ratio at the *Igll1* promoter and TSS. (**D**) 3D DNA-FISH to monitor the position of *Igll1* alleles (green) relative to γ-satellite DNA (red, blue is DAPI). The percentage of *Igll1* alleles associated with γ-satellite DNA is shown as mean ± SE. Where indicated, cells were treated over night with TSA (1 ng/ml) and/or 4-OHT. At least 300 *Igll1* alleles were scored for each experimental condition across 3 independent biological replicates. The impact of TSA was statistically significant across replicates (p=9.54 × 10-18 GLM binomial logit).
The following source data is available for figure 6:

**Source data 1.** Numerical data used to generate *Figure 6A,B,C and D*.

remodeled *Igll1* promoter even after the binding of CHD4 and MBD3 declined, which may in part explain why promoter accessibility was not re-instated, suggesting that Ikaros-mediated NuRD recruitment and eviction of BRG1 may cooperate in the repression of Ikaros target genes.

Interestingly, our high-resolution view of Ikaros-mediated gene silencing indicates a temporal dissociation of enzymatic activities associated with NuRD where histone deacetylation occurred after chromatin remodelling and transcriptional repression (*Figure 7D*). CHD4, but not HDAC activity was required for reduced promoter accessibility, the loss of RNAP2, and transcriptional repression. Although dispensable for the initiation of transcriptional repression, HDAC activity did facilitate stable transcriptional silencing, nucleosome occupancy of target promoters, and locus repositioning to pericentromeric heterochromatin.

The silencing mechanisms delineated here are interdependent, as illustrated by the role of HDACs and the involvement of CHD4 in competition between Ikaros and EBF1 for access to the *Igll1* promoter, RNAP2 occupancy and locus repositioning to repressive nuclear environments, a hallmark of heritable gene silencing (*Brown et al., 1999*; *Su et al., 2004*). Our data link locus repositioning to chromatin state by demonstrating that both histone deacetylation and CHD4-mediated chromatin remodeling contribute to efficient repositioning of the *Igll1* locus. Histone deacetylation and nucleosome positioning likely contribute to the compaction of the locus, facilitating 'like-with-like' associations of chromatin states (*Jost et al., 2014*).

Genes that are induced rapidly in response to TLR signaling in macrophages are often regulated at the level of transcriptional elongation (*Hargreaves et al., 2009*) and typically do not require ATP-dependent chromatin remodelling (*Ramirez-Carrozzi et al., 2009*, *2006*). At the Ikaros target genes *Igll1* and *Myc* as well as genome-wide, RNAP2 recruitment, rather than RNAP2 elongation, was the primary mechanism by which Ikaros repressed RNAP2-dependent transcription. The role of RNAP2 was not to keep nucleosomes away from the *Igll1* and *Myc* promoters because experimental depletion of RNAP2 did not result in a loss of promoter accessibility. Experimental depletion of RNAP2 and the resulting loss of RNAP2-mediated transcription did not interfere with Ikaros-induced promoter remodelling. This is consistent with a direct role for Ikaros in target gene repression, and reminiscent of primary response gene induction in the absence of protein synthesis. With respect to the role of active chromatin remodeling, however, Ikaros-mediated repression was unlike the induction of primary response genes, and reminiscent of nuclear hormone receptor signaling where several enzymes cooperate to remodel chromatin within minutes of hormone addition (*Ballaré et al., 2013*; *Grøntved et al., 2013*; *Nacht et al., 2016*; *Vicent et al., 2011*).

The molecular determinants of whether IKAROS binding results in gene activation or repression remain to be explored. For genes that are repressed by Ikaros with fast kinetics, we favour a model where transcriptional repression by IKAROS involves active, CHD4-mediated chromatin remodeling of the *Igll1* and *Myc* target gene promoters. This restricts promoter access, excludes RNAP2 and the transcriptional activator EBF1, and ultimately limits binding of Ikaros itself. The emerging picture is that the rapid repression of developmentally regulated genes utilizes complex and interdependent mechanisms, and that complexity does not come at the expense of speed in transcriptional state transitions.

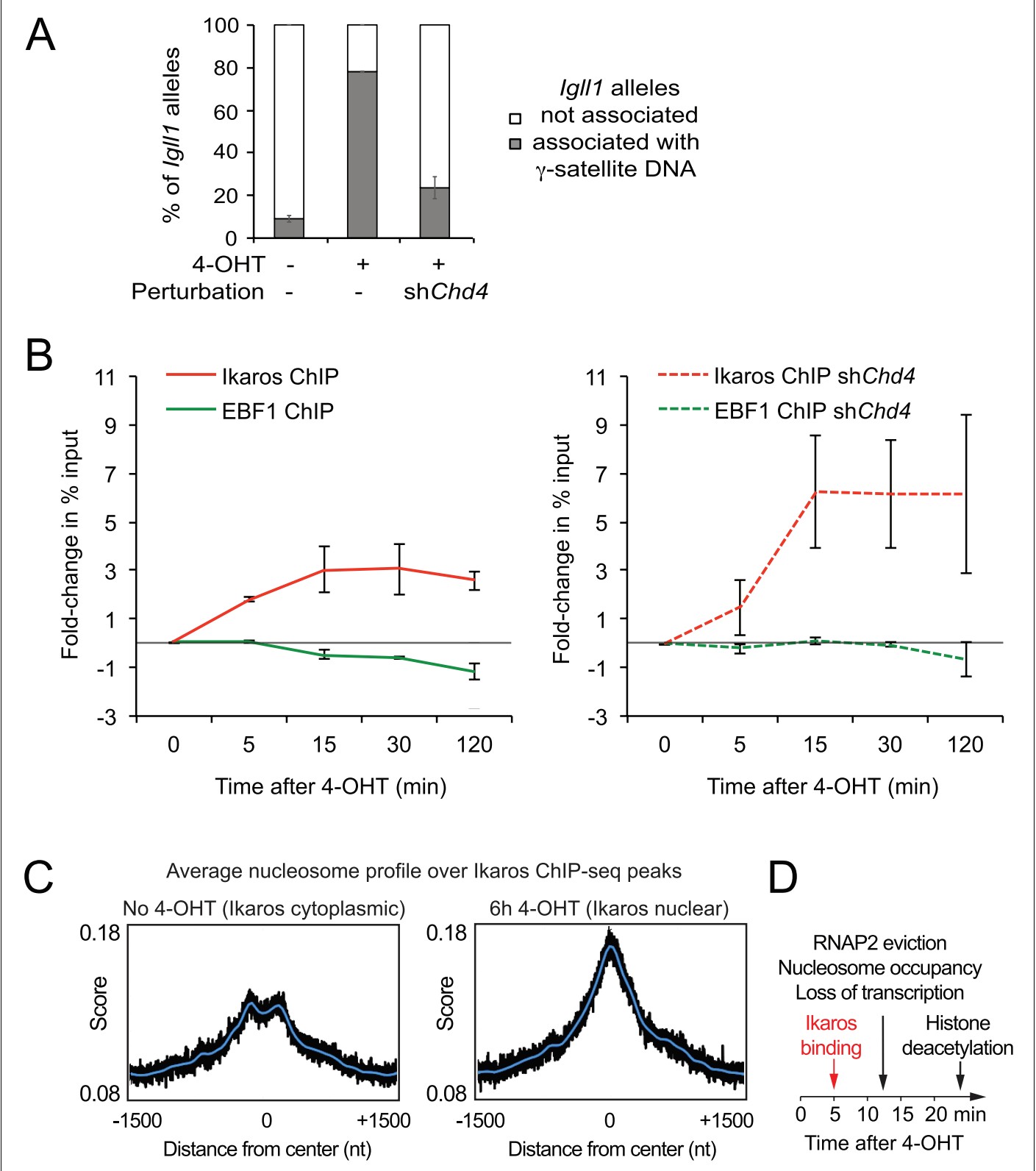

**Figure 7.** Interdependence of silencing mechanisms leveraged by Ikaros. (**A**) 3D DNA-FISH to monitor the position of *Igll1* alleles (green) relative to γ-satellite DNA (red, blue is DAPI). The percentage of *Igll1* alleles associated with γ-satellite DNA is shown as mean ± SE. Where indicated, control or sh*Chd4* cells were treated over night with 4-OHT. At least 300 *Igll1* alleles were scored for each experimental condition across 3 independent biological replicates. The impact of *Chd4* knockdown was statistically significant across replicates (p=5.54×10-38 GLM binomial logit). (**B**) ChIP kinetics of Ikaros

*Figure 7 continued on next page*

*Figure 7 continued*

and EBF binding to the *Igll1* promoter in control (left) and sh*Chd4* cells (right). Increased binding of Ikaros to the *Igll1* promoter was significant for both control and sh*Chd4* cells, decreased binding of EBF1 was significant in control, but not in sh*Chd4* cells. Mean ± SE, 3 independent biological replicates. Ikaros and EBF1 binding at 15, 30 and 120 min were significantly higher in sh*Chd4* than control cells. (C) MNase-seq data from 3 independent biological replicates were integrated with Ikaros ChIP-seq data to show nucleosome occupancy at Ikaros binding peaks before and 6 hr after nuclear translocation of Ikaros. (D) Dynamics of Ikaros binding, RNAP2 eviction, loss of primary transcripts, nucleosome invasion, and histone deacetylation.

The following source data is available for figure 7:

**Source data 1.** Numerical data used to generate *Figure 7A and B*.

## Materials and methods

### Cell culture

The B3 cell line was generated and characterised in our lab (*Brown et al., 1997*). B3 and primary pre-B cells were maintained in Iscove's Modified Dulbecco's Medium (IMDM, Invitrogen), 10% FCS, penicillin and 100 µg/ml streptomycin) (GIBCO, Invitrogen). For primary pre-B cells, media were supplemented with $\beta$-mercaptoethanol, L-glutamine 100 U/ml, 5 ng/ml recombinant mouse IL-7 (R and D Systems) and ST-2 feeder cells. HA-Ikaros-ERt2-IRES-GFP or -Cherry in the MSCV retroviral vector was modified by insertion of a TEV protease recognition sequence to yield HA-Ikaros-TEV-ERt2-IRES-GFP or -Cherry and where indicated B3 cells were co-transfected with split-TEV constructs where the N- and C-termini of TEV protease were fused to FRB (FLAG-FRB-N_TEV- IRES-dsRED) and FKBP (FLAG-FKBP-C_TEV-IRES-CFP), respectively. Transduced cells were sorted by flow cytometry for homogeneous expression of constructs. MSCV HA-Ikaros-IRES-GFP and IRES-GFP control vectors, virus production and spin infection have been described.

Nuclear translocation of Ikaros-ERt2 was induced by Ikaros was induced by 0.5 µM 4-hydroxyta-moxifen (4-OHT, Sigma-Aldrich) and TEV activity was induced by rapamycin (25 nM, Selleckchem). Nuclear translocation was monitored by anti-HA (Covance MMS-101R) immunofluorescence staining and confocal microscopy as described and images were processed using Leica and Image J software.

Where indicated, cells were treated with 1 µM triptolide (Sigma), 1 ng/ml trichostatin A (TSA, Sigma) or 10 µM MS-275 (Selleckchem).

shRNA for *Chd4* (5'-GACTACGACCTGTTCAAGCAG-3') and pQsupR control plasmid were used as advised by the manufacturer (Invitrogen) and media were supplemented with 20 µM pan caspase inhibitor (R and D Systems, FMK001).

### Quantification of mRNA and protein expression

mRNA extraction, reverse transcription PCR and immunoblotting was as described. We used antibodies to histone H3 (Abcam AB1791), acetylated histone H3 (Abcam AB47915), acetylated histone H4 (Millipore 06–866), RNAPII (Santa Cruz sc-899), TFIIB (Santa Cruz sc-225), CHD4 (Abcam AB72418), and tubulin (Sigma T9026).

### Chromatin accessibility by MNase and chromatin immunoprecipitation (ChIP)

MNase digestion was done as described (*Carey and Smale, 2007*; *Henikoff et al., 2011*). Cells were cross-linked for 10 min at room temperature in 1% formaldehyde in 10% fixation buffer (0.5 mM EGTA pH 8.0, 100 mM NaCl, 1 mM EDTA, 50 mM HEPES pH 8.0, 10% formaldehyde). Fixation was stopped by glycine (140 mM). Cells were washed twice with cold PBS (4°C, 5 min, 900rcf) and resuspended in lysis buffer (5 mM PIPES pH 8.0, 85 mM KCl, 0.5% NP-40) supplemented with protease inhibitor cocktail (Roche), 0.15 mM spermine (Sigma), and 0.5 mM spermidine (Sigma) on ice for 20 min. Sample were washed and resuspended in MNase CaCl$_2$+ buffer (10 mM Tris-HCl, pH 7.4, 15 mM NaCl, 60 mM KCl, 0.15 mM spermine, 0.5 mM spermidine) and digested with MNase (200 U/ml, Worthington) at in room temperature for 10 min. Digestion was stopped by 20 µl 100 mM EDTA, 10 mM EGTA per $2 \times 10^6$ cells. As a control, an equal number of cells were treated in parallel without

adding MNase. Samples were reverse crosslinked, DNA purified by phenol/chloroform and quantified (PicoGreen dsDNA quantitation kit, Life Technologies).

For MNase-qPCR, 2 ng DNA were used per qPCR reaction. Results were normalised to undigested controls and compared to the inactive *Rex1* or *αActin* promoter.

For MNase-seq, 50 ng DNA after MNase digestion (carrier ethanol treatment for 6 hr or 0.5 μM 4-OHT treatment for 6 hr) were used to prepare MNase-seq samples (Next Ultra, NEB) without size selection steps. Libraries from three biological replicates were sequenced (Illumina Hi-seq 2500, 50 bp paired end) to ~500–600 M reads per condition. Reads were aligned to mouse genome mm9 and reconstructed into complete fragments in silico. To calculate normalised coverage per base for MNase data, fragments with length between 110 bp and 170 bp were normalised to total reads outside of mm9 blacklisted regions.

Ikaros ChIP was as described using a C-terminal Ikaros antibody provided by Dr Stephen Smale that has been extensively characterised in ChIP-seq experiments (*Ferreirós-Vidal et al., 2013*; *Bossen et al., 2015*). We used an EBF1 antibody (Santa Cruz sc-137065) previously characterised by ENCODE for EBF1 ChIP-seq. For RNAP2, chromatin remodeler and histone modification ChIP (*Brookes et al., 2012*) cells were fixed as described for MNase, lysed in 25 mM HEPES, 1.5 mM MgCl2, 10 mM KCl, 0.1% NP-40, pH 7.9 at $50 \times 10^6$ cells/ml on ice for 15 min, centrifuged at 900 rcf for 10 min at 4°C, resuspended in 140 mM NaCl, 50 mM HEPES, 1 mM EDTA, 1% Triton X-100, 0.1% SDS, 0.1% sodium deoxycholate, pH 7.9 and sonicated for 25 min at 4°C with 30 s on, and 30 s off (Bioruptor, Diagenode). Sonicated chromatin was recovered as supernatant after centrifugation at 25,000 rcf for 15 min at 4°C and incubated with antibodies to histone H3 (Abcam AB1791), acetylated histone H3 (Abcam AB47915), acetylated histone H4 (Millipore 06–866), RNAP2 (Santa Cruz sc-899), TFIIB (Santa Cruz sc-225), CHD4 (Abcam AB70469), MBD3 (Bethyl A302-528A) or Brg1 (Millipore, 07–478) on a rotating platform at 4°C overnight, washed sequentially in 140 mM NaCl, 50 mM HEPES, 1 mM EDTA, 1% Triton X-100, 0.1% SDS, 0.1% sodium deoxycholate, pH 7.9 followed by 500 mM NaCl, 50 mM HEPES, 1 mM EDTA, 1% Triton X-100, 0.1% SDS, 0.1% sodium deoxycholate, pH 7.9 followed by 250 mM LiCl, 50 mM HEPES, 1 mM EDTA, 0.5% NP-40, 0.5% sodium deoxycholate, pH 8.0 and finally 10 mM Tris-HCl, 1 mM EDTA, pH 7.5. Reverse crosslinked DNA was purified by phenol-chloroform extraction. For ChIP-seq, 10 ng ChIP and input DNA were used to prepare libraries (Next Ultra, NEB) with size selection for fragments between 100 and 300 bp. After 50 bp single-end sequencing (Hi-seq 2500, Illumina, ~155 M reads per condition), reads were aligned to mm9 using BWA version 0.7.5. SAM files were generated, aligned, sorted and indexed with samtools version 0.1.18. Duplicate reads were marked using Picard MarkDuplicates. The quality of the datasets and predicted fragment lengths were analysed using the ChIPQC Bioconductor package. Normalised BigWig files were generated using rtracklayer (version 1.3.2) with signal normalised to total mapped fragments per sample. ChIP-seq sample quality was assessed using ChIPQC (version 1.8.7) (*Carroll et al., 2014*). For the analysis of differential polymerase signal in genes and TSS regions, gene models for mm9 were retrieved from Ensembl Biomart. Total reads were counted within TSSs and gene bodies and analysis of differential binding performed using DEseq2 with efficiency and library complexity derived from ChIPQC supplied to the DEseq2 model as normalisation factors (*Love et al., 2014*). MNase signal over Ikaros ChIP-seq peaks (*Ferreirós-Vidal et al., 2013*) was calculated using the Bioconductor soGGi package (*Dharmalingam and Carroll, 2015*) and differential MNase signal identified using the DChIPrep package (*Chabbert et al., 2016*). Differential gene expression was based on a p-value of <0.05 after Benjamini-Hochberg correction for multiple testing (adj. p<0.05).

## RT-PCR primers

| Target | Forward | Reverse |
|---|---|---|
| *Igll1* | GTGTCCACCACATACTTTCCCCA | CACTCATTCTAGCCTCTAGTCCGTG |
| *Myc* | TTCTCACCTGTGCCCTAACC | GGTTTGCCTCTTCTCCACAG |
| *Zfp36* | GCTGGCTGGAAATGAGAGAG | CCCCCTACCTCAACCTTAGC |

## MNase-PCR and ChIP-PCR primers

| Target | Forward | Reverse |
|---|---|---|
| *Igll1* Prom. short | CTGTGAGTGAAAACAGTTAGGCTTGC | ACCAGCAGGCACACCCCAGTG |
| *Igll1* Prom. long | CAAACCCCAGGCTGTCTCTA | GGCAGCTGTGAGTGAAAACA |
| *Igll1* TSS short | GGTGGAAACTAGAGACAGCCTGG | CGGCAAAAGGATTGTTCCTCC |
| *Igll1* TSS long | TGGCCAAAAGCTATTCCAGT | GGGTAGTCCCTTTGGGAGAG |
| *Igll1* +200 nt | TGCTGTTGGGTCTAGTGGATGG | GCCTGTTGCTTCCCACTGAAG |
| *Igll1* +1.3 kb | CAGCCAGTATCCCGACAAGT | AGAATCTGCTGGGCCTGATA |
| *Igll1* TTS | ACTGGGTTCCATGACTCCAC | TCACTGTCTTTCCTGTGCCTAA |
| *Myc* Prom. | CTCACTCAGCTCCCCTCCT | CTCCCCTCCCTTCTTTTTCC |
| *Myc* TSS | AGGGATCCTGAGTCGCAGT | CGCTCACTCCCTCTGTCTCT |
| *Myc* +350 nt | CCTAAGAAGGCAGCTCTGGA | GCTGATGTTGGGTCAGTCG |
| *Myc* +1 kb | AACCAGAGGGAATCCTCACA | GAACCGCTCAGATCACGACT |
| *Myc* TTS | GCCCAGTGAGGATATCTGGA | ACCGCAACATAGGATGGAGA |
| *Zfp36* Prom. | CATGCAAAATGTGCCTGAAC | CGCTACCATCACCTCCAGTT |
| *Zfp36* TSS | GAATGGCCTTGGTGAAGAGA | GCCCCATAAAAGGAGAAAGC |
| *Zfp36* +200 nt | TACGCGAGTGACAGCAGTGT | GCTGGCTGGAAATGAGAGAG |
| *Zfp36* +0.6 kb | CACTTCACGAGCTTGCCAGT | AGTCAGGTTCTCCCTGGAGTT |
| *Zfp36* TTS | GGCTGGTAACGTCACTTCCT | GGTGGGTAAACTGACCTCCA |

To assess the impact of nuclear Ikaros on Ikaros-EBF1 competition, we implemented kinetic models of increasing complexity. First, we considered a 2-state model where DNAE and DNAI correspond to DNA bond by EBF1 and Ikaros with the binding rate constants are kE and kI (*Equation 1*).

$$[DNA_E] + Ikaros \rightleftharpoons [DNA_I] + EBF1 \tag{1}$$

The corresponding rate equations are:

$$\begin{cases} \frac{d[DNA_E]}{dt} = -k_I[Ikaros][DNA_E] + k_E[EBF1][DNA_I] \\ \frac{d[DNA_I]}{dt} = k_I[Ikaros][DNA_E] - k_E[EBF1][DNA_I] \end{cases} \tag{2}$$

To solve these rate equations by numerical integration we used KinTek Explorer (KinTek Corporation, Austin, TX; *Johnson, 2009*; *Figure 1D*). The 2-state model in *Equation 2* cannot recapitulate the features observed in kinetic ChIP experiments (*Figure 1B*) using the estimated parametes Ikaros $k_{on}$ 0.3 $k_{off}$ 0.052; EBF1 $k_{on}$ 0.036 $k_{off}$ 0.033, or any other parameters tested. We therefore implemented a 3-state model that includes a population of free (unbound) DNA (*Equation 3*).

$$[DNA_E] \rightleftharpoons [DNA] \rightleftharpoons [DNA_I] \tag{3}$$

In this model, Ikaros and EBF1 can compete for the same binding site only if it is free. The EBF1 and Ikaros dissociation rate constants are k-E and k-I, respectively. The corresponding rate equations are in *Equation 4*.

$$\begin{cases} \frac{d[DNA_E]}{dt} = -k_{-E}[DNA_E] + k_E[EBF1][DNA] \\ \frac{d[DNA]}{dt} = k_{-E}[DNA_E] + k_{-I}[DNA_I] - k_E[EBF1][DNA] - k_I[Ikaros][DNA] \\ \frac{d[DNA_I]}{dt} = k_I[Ikaros][DNA] - k_{-I}[DNA_I] \end{cases} \tag{4}$$

Parameters were estimated as Ikaros $k_{on}$ 0.3 $k_{off}$ 0.052; EBF1 $k_{on}$ 0.036 $k_{off}$ 0.033, start conditions: I-DNA 0.1, E-DNA 0.1, free DNA 0.8, similar results were obtained with a range of parameters. The 3-state model recapitulates the primary features observed in the ChIP experiments (*Figure 1B,E*), but does not account for the gradual increase in nuclear Ikaros observed experimentally (*Figure 1A*

and *Figure 1—figure supplement 1B*) or the possibility of binding site occupancy by nucleosomes (non-accessible DNA or DNAN). We introduced DNAN in a 4-state model (*Equation 5*) where kN and k-N are the nucleosome binding and dissociation rate constants, respectively:

$$\begin{cases} \frac{d[\text{DNA}_E]}{dt} = -k_{-E}[\text{DNA}_E] + k_E[\text{EBF1}][\text{DNA}] \\ \frac{d[\text{DNA}]}{dt} = k_{-E}[\text{DNA}_E] + k_{-I}[\text{DNA}_I] + k_{-N}[\text{DNA}_N] \\ \quad - k_E[\text{EBF1}][\text{DNA}] - k_I[\text{Ikaros}][\text{DNA}] - k_N[\text{Nucleosome}][\text{DNA}] \\ \frac{d[\text{DNA}_I]}{dt} = k_I[\text{Ikaros}][\text{DNA}] - k_{-I}[\text{DNA}_I] \end{cases} \tag{5}$$

We used the parameters Ikaros $k_{on}$ 0.3 $k_{off}$ 0.052; EBF1 $k_{on}$ 0.036 $k_{off}$ 0.033; nucleosome $k_{on}$ 0.13 $k_{off}$ 0.003, start conditions: I-DNA 0.1, E-DNA 0.1, free DNA 0.8, starting concentration of Ikaros: 0.1, Ikaros increment per time interval: 2.0 and obtained the kinetic curves shown in *Figure 1—figure supplement 1C*. We experimentally tested the model predictions in kinetic ChIP experiments. Data from control cells (*Figure 7B*, left) correspond well to the 4-state model in *Equation 5*, suggesting a role for nucleosomes in restricting binding of Ikaros and EBF1. Ikaros and EBF1 binding were increased in the absence of the nucleosome remodeler CHD4 (*Figure 7B*, right), in line with predictions by the 3-state model in *Equation 4* without DNAN.

## Acknowledgements

We thank Drs Jorja and Steven Henikoff for help with MNase-seq, Drs John Schwabe and Ross Chapman for helpful discussion, the STATegra consortium and in particular Drs David Gomez-Cabrero, Peri Noori, Sunjay Fernandes, Mathilda Eriksson and Jesper Tegner for sharing RNA-seq data prior to publication, Drs Pierangela Sabbatini and Niall Dillon for *Igll1* cosmids, Dr James Elliott and Thomas Adejumo for cell sorting, and Dr Laurence Game and her team for sequencing. This work was funded by the Medical Research Council UK, Wellcome, and UK-China Scholarships for Excellence (a scheme run jointly by the Department for Business, Innovation and Skills and the China Scholarship Council).

## Additional information

### Funding

| Funder | Grant reference number | Author |
|---|---|---|
| UK-China Scholarships for Excellence | | Ziwei Liang |
| Wellcome | 098021/Z/11/Z | Brian Hendrich |
| Medical Research Council | MC-A652-5PY20 | David Rueda<br>Amanda G Fisher<br>Matthias Merkenschlager |
| Wellcome | 099276/Z/12/Z | Matthias Merkenschlager |

The funders had no role in study design, data collection and interpretation, or the decision to submit the work for publication.

### Author contributions

ZL, Conceptualization, Investigation, Methodology, Writing—original draft; KEB, Investigation; TC, Data curation, Formal analysis; BT, IFV, Investigation, Methodology; BH, Methodology; DR, Conceptualization, Formal analysis; AGF, Conceptualization, Writing—review and editing; MM, Conceptualization, Formal analysis, Supervision, Writing—original draft, Writing—review and editing

### Author ORCIDs

Benjamin Taylor, http://orcid.org/0000-0001-6101-3786
Brian Hendrich, http://orcid.org/0000-0002-0231-3073
Matthias Merkenschlager, http://orcid.org/0000-0003-2889-3288

## Additional files

### Major datasets

The following dataset was generated:

| Author(s) | Year | Dataset title | Dataset URL | Database, license, and accessibility information |
|---|---|---|---|---|
| Liang Z, Carroll T, Merkenschlager M | 2017 | A high-resolution map of transcriptional repression by Ikaros | https://www.ncbi.nlm.nih.gov/geo/query/acc.cgi?acc=GSE89716 | Publicly available at the NCBI Gene Expression Omnibus (accession no: GSE89716) |

The following previously published dataset was used:

| Author(s) | Year | Dataset title | Dataset URL | Database, license, and accessibility information |
|---|---|---|---|---|
| Bossen C, Murre CS, Chang AN, Mansson R, Rodewald H, Murre C | 2015 | Brg1 activates enhancer repertoires to establish B cell identity and modulate cell growth | https://www.ncbi.nlm.nih.gov/geo/query/acc.cgi?acc=GSE66978 | Publicly available at the NCBI Gene Expression Omnibus (accession no: GSE66978) |

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
