## [Decision Letter]

Thank you for submitting your article "A high-resolution map of transcriptional repression" for consideration by *eLife*. Your article has been reviewed by three peer reviewers, one of whom, David N Arnosti (Reviewer #1), served as Guest Editor, and the evaluation has been overseen by Jessica Tyler as the Senior Editor.

The reviewers have discussed the reviews with one another and the Reviewing Editor has drafted this decision to help you prepare a revised submission.

Summary:

Three reviewers have read your manuscript, and we agree that this work is potentially suitable for publication in *eLife*, after you have addressed points raised in the review process.

Essential revisions:

1) Data reproducibility. A major point raised in the review process was whether the ChIP data shown represents biologically independent measurements or technical replicates from single preparations of chromatin. A number of experiments (Figure 2, Figure 3, Figure 4, Figure 5, Figure 6) show detailed measurements about protein occupancy or histone modification, but in some cases, the authors do not provide enough information about the data presented to allow the reader to judge if the changes measured are convincing demonstrations of biological processes. Specifically, for most of the ChIP measurements, the figure legend notes n=3; is that three measurements of single chromatin preparation, or three independent preparations, independently measured? Certain experiments (e.g. 4D) are described as "independent experiments", thus, the former cases may represent a single biological experiment (chromatin prep), with independent chromatin IP measurements. One line of evidence supporting this view is that individual figures have fairly tight error bars, yet vary quite a bit between experiments, e.g. RNAP drop in Figure 3 is 3-5 fold, while in Figure 4 the change is ~2 fold. The paper should clearly distinguish technical from biological replicates, and indicate if the results are supported by more than one experiment. Clearly, the strength of the conclusions is related to the reproducibility of the effects from separate observations.

2) Relationship of findings to prior knowledge about Ikaros. A second point from the review was need to set the authors' findings in a more general context, to show how information about HDAC, NuRD, and chromatin structure in regulation of Myc and Ipll1 supports or contradicts previous studies on Ikaros action. For instance, earlier work from Georgopoulos showed that on the CD4 intronic enhancer, Ikaros and NuRD appear to antagonize each other's activity, such that loss of CHD4 results in enhanced, not reduced Ikaros, activity. Similarly, previous studies have implicated basal factor pTefb-Ikaros interactions, as well as polycomb. Other studies have shown concomitant binding of Ikaros and Ebf1 at the Igll1 promoter in large pre-B cells where the gene is expressed, raising the question of whether Ikaros activates in this context. The paper would be much more valuable to the community if the authors integrate their findings better with previous work, and in the Discussion propose how similarities or differences are to be reconciled. As it stands, the main summary concluding the paper is rather superficial.

3) Presentation and use of genome-wide data. The reviewers found that the integration of genome-wide information was potentially useful but incomplete; the quality of genome-wide information should be indicated with standard statistical measures (e.g. how much agreement is found among the triplicate MNase experiments? reproducibility of peaks etc.), even if the analysis of the entire dataset is being prepared in a separate study. MNase digestion patterns at two loci discussed, but overall significance is difficult to assess; is variation indicative of a change in nucleosome position, occupancy or other aspect of the individual experiments? It is not clear if the RNA pol II genome-wide study was similarly performed in triplicate. A related point to consideration of genome-wide data is related to point 2; it is reported that out of the 924 genes repressed upon increase in Ikaros expression, 372 show a reduction in RNAP2. How are the other 550 genes regulated? Are these not directly repressed by the Ikaros-NuRD complex? It would help this paper to provide enough information to understand Table 1 and Supplementary Data Table 1, which enumerate and list genes affected by Ikaros. For instance, the two genes studied in detail here are direct targets of Ikaros, and show a loss of RNA Pol II. A large number, but not the majority of other repressed genes also show a loss of Pol II; which of these are expected to be direct targets? An additional point regarding the table is the lack of clarity of how genes are assigned to it – is for example a log2 value of 0.05 even a significant change?

4) Modeling. The reviewers had differing opinions about the modeling; some asked for further validation through additional measurements of actual nucleosome positioning and Ikaros occupancy at individual sites, while another view was that the models are useful enough to point toward future studies of mechanism. A justification of the application of the modeling would strengthen this aspect of the study. In addition, a clarification of the actual mechanism suggested is requested: Ikaros has been variously described as a recruiter of NuRD, or an antagonist, or binding in an overlapping manner at promoters but not enhancers. The kinetics of binding shown in 4D indicate that a model with simple, stable recruitment of NuRD by Ikaros is too simple. But it is not clear what mechanism the authors are proposing, based on their findings.

5) Endogenous Ikaros vs. induced form. An increase in Ikaros nuclear localization is assumed from the immunofluorescence data but this is not a very quantitative method. A western blot analysis for Ikaros in the nuclear vs, cytoplasmic fractions at the different time points should be performed. The way the ChIP data is presented, it is not clear how much endogenous Ikaros is already present at the target promoters before the induction of the tamoxifen-regulated protein, and how this compares with the induced levels. Previous studies have already noted that Ikaros is present at these genes, so how much of the modifications are due to overexpression of the protein? Why would nucleosome changes results only from overexpression; is this a function of rapid interconversion between active and inactive states on the genes with only endogenous Ikaros? In addition, the paper is silent about the Ikaros-like Aiolos protein; is it present at these genes, and does it play a role in regulation under these conditions?

6) MNase interpretation. The purported identification of "fragile" sites by MNase tests was not viewed as very strong, for a number of reasons. To identify such changes as an alternatively-bound form of the nucleosome, one would wish for a titration of MNase, not just a single digestion. In addition, it would be important to know if the change is due to a nucleosome shifting in position, rather than being weakly bound. Finally, the differential amplification in 6B with "short" and "long" amplicons may or may not be significant – it is not clear if the n=3 is a technical replicate, in which case, the support is weak.

A number of technical points relating to reagents, methods, and data presentation should be addressed:

7) The antibodies for Ikaros and EBF1 are not described; in the Materials and methods, a reference is made to Ferreiros-Vidal 2013, but EBF1 is not used there. What is the measure of specificity for these reagents? Is there other data to support the EBF1 binding shown here?

8) In Figure 1, "enrichment" is the measure of chromatin binding for Ikaros and EBF1. The Materials and methods don't describe how these calculations were performed. One way might be to measure the signal at some non-bound site, and divide the ChIP signal% input by this background control. If so, what is their defined background? What were the% input values for these promoters and other locations?

9) For Pol II and TFIIB measurements, the manuscript shows directly the% input recovered from ChIP, which is a preferable approach for chromatin aficionados (their values ~1% are quite credible). The Y-axis is labeled "enrichment", however, which is confusing, because presumably the% input was not normalized to signal at another locus.

10) In Figure 3—figure supplement 1, part B, using the TEV-activated Ikaros system, depletion of Pol II does further affect MNase sensitivity, which is different from the system with 4-OHT induced Ikaros. The authors don't comment on this, though they previously make the point that Pol II eviction is not causing the changes in MNase sensitivity.

11) Figure 5—figure supplement A: TSA treatment apparently induces a decrease in histone acetylation levels? Please comment.

12) The differences for pericentric chromatin localization +/- TSA (6D) and loading of Ikaros with or without CHD4 (7B) may or may not be significant; it is not clear how reproducible these differences are in independent experiments. (See Point 1). In 6D, apparently two experiments were conducted; is this data from one?

13) It is not clear why the KinTec modeling used the specific parameters indicated in Materials and methods. Have these been measured in a particular system?

14) In subsection “Loss of RNAP2, reduced promoter accessibility, and transcriptional repression are early and near-simultaneous events” the authors state that they use primer pairs that span introns as a means to measure unspliced transcripts. They presumably mean primers that span individual exon/intron junctions? The primers used should be included in Materials and methods.

[Editors' note: further revisions were requested prior to acceptance, as described below.]

Thank you for resubmitting your work entitled "A high-resolution map of transcriptional repression" for further consideration at *eLife*. Your revised article has been favorably evaluated by Jessica Tyler as Senior editor and a Reviewing editor.

The manuscript has been improved but there are four remaining issues that need to be addressed before acceptance, as outlined below:

In this revision of Liang et al., the authors have addressed the points raised in the first round of review. Regarding data reproducibility, the authors have indicated in figure legends which datasets are averages of multiple biological experiments, and which are representative. Additional experiments were performed for 3D FISH, and a Figure 2—figure supplement 1 shows the reproducibility of ChIP and MNase experiments. Regarding previous observations about Ikaros action in different cell types and promoters, they clarify that their findings for c-myc and Igll1 promoters may represent just one side of this protein's regulatory activities, which includes activation at Zfp36.

The reviewers had asked for more specific analysis of genome-wide gene expression, since only a minority of genes showed the Pol II decrease found at the two promoters of interest. In the response to the reviewers, the authors reanalyze the data, and conclude that for a majority of promoters, it appears that Pol II decrease at the promoter is the trend, and that differences in sensitivity of measuring mRNA vs. Pol II occupancy may explain some of the discrepancy.

1) In addition to having this information in the letter of response, the manuscript should indicate that they believe the majority of genes are showing reductions in Pol II, but that differential sensitivity may be an issue. Otherwise it seems the authors are content to ignore what might be a mechanism that impacts the majority of repressed genes.

The use and interpretation of the models were questioned; in response, the authors provide more information on modeling, and emphasize that their many biological measurements point away from the simple two-state model, where Ikaros directly dislodges EBF1. This conclusion seems to be strongly supported by the data. They emphasize the failure to find reasonable parameters for the simple model, rather than over-interpreting specific parameters for more complex models, which seems a reasonable conclusion.

The nature and quality of experiments from genome-wide experiments is better explained and illustrated by the additional supplemental figure noted above, by correlation analyses in the response letter, and data deposited in GEO.

In response to the question about how endogenous Ikaros figures into this system, the authors note that they find similar responses in CRISPR'd cells lacking endogenous Ikaros, and show preliminary data in the letter. As asked, they also carry out a Western blot showing the levels of native and induced Ikaros over the time course, showing that the levels of the inducible form is approximately the same as that of the endogenous protein.

2) The Western blot data should be included in the manuscript as part of, or attached to, Figure 1, where the system is introduced.

The interpretation of a "fragile" nucleosome state was questioned, based on the limited probing with one concentration of MNase. The authors carry out a titration, showing a similar trend, but also soften their conclusion, noting that it appears that it is either nucleosome depletion or movement that is affecting the repressed promoter.

A number of minor points were addressed including provenance of antibodies, figure labeling, and explanation in the figure legend that nascent RNA was measured with intron-exon boundary spanning primers.

3) The text still refers to "intron-spanning primers" – that needs to be fixed.

4) Typo Figure 7 Average nuceosome profile

---

## [Author Response]

Essential revisions:

1) Data reproducibility. A major point raised in the review process was whether the ChIP data shown represents biologically independent measurements or technical replicates from single preparations of chromatin. A number of experiments (Figure 2, Figure 3, Figure 4, Figure 5, Figure 6) show detailed measurements about protein occupancy or histone modification, but in some cases, the authors do not provide enough information about the data presented to allow the reader to judge if the changes measured are convincing demonstrations of biological processes. Specifically, for most of the ChIP measurements, the figure legend notes n=3; is that three measurements of single chromatin preparation, or three independent preparations, independently measured? Certain experiments (e.g. 4D) are described as "independent experiments", thus, the former cases may represent a single biological experiment (chromatin prep), with independent chromatin IP measurements. One line of evidence supporting this view is that individual figures have fairly tight error bars, yet vary quite a bit between experiments, e.g. RNAP drop in Figure 3 is 3-5 fold, while in Figure 4 the change is ~2 fold. The paper should clearly distinguish technical from biological replicates, and indicate if the results are supported by more than one experiment. Clearly, the strength of the conclusions is related to the reproducibility of the effects from separate observations.

The number of experiments refers to independent biological replicates in all cases. Everything from cells/chemical treatment, chromatin/RNA isolation/digestion, ChIP/reverse transcription/PCR, library preparation and sequencing was done in the form of independent repeats on different days for the number of times indicated in the figure legends. We have added the term “independent biological replicates” to the revised figure legends to indicate this fact. The data are highly reproducible, as documented in the new Figure 2—figure supplement 1 and the source data files supplied for each figure.

The difference between the drop in RNAP2 in Figure 3 to 5-fold) versus Figure 4 (~2-fold) reflects the expression of IKAROS-IRES-GFP in 3B and C versus IKAROS-IRES-Cherry in Figure 4 where the cells were transduced with both inducible IKAROS and shRNA plasmids and expressed lower levels of IKAROS. Importantly, however, the responses obtained were consistent, as indicated by the SE from 3 independent biological replicates in Figure 4 and C.

2) Relationship of findings to prior knowledge about Ikaros. A second point from the review was need to set the authors' findings in a more general context, to show how information about HDAC, NuRD, and chromatin structure in regulation of Myc and Ipll1 supports or contradicts previous studies on Ikaros action. For instance, earlier work from Georgopoulos showed that on the CD4 intronic enhancer, Ikaros and NuRD appear to antagonize each other's activity, such that loss of CHD4 results in enhanced, not reduced Ikaros, activity. Similarly, previous studies have implicated basal factor pTefb-Ikaros interactions, as well as polycomb. Other studies have shown concomitant binding of Ikaros and Ebf1 at the Igll1 promoter in large pre-B cells where the gene is expressed, raising the question of whether Ikaros activates in this context. The paper would be much more valuable to the community if the authors integrate their findings better with previous work, and in the Discussion propose how similarities or differences are to be reconciled. As it stands, the main summary concluding the paper is rather superficial.

The paragraph above raises several important issues, and we have taken the liberty to structure it into 5 different points (i) to (v).

i) The data reported in our manuscript do not in our view contradict previous reports, largely because no previous reports have examined the impact of IKAROS and NuRD on the time scale investigated here. However, it is worth noting that previous reports occasionally contradict each other. For example, there is marked disagreement between the Georgopoulos (Zhang et al. Nat Immunol. 2011, 13: 86-94) and Busslinger labs (Schwickert et al., 2014). We did not examine CHD4 positioning in Ikzf1-deficient cells, nor did we investigate the role of IKAROS, HDACs, or CHD4 at the proximal Cd4 enhancer (Williams et al., Immunity. 2004, 20: 719-33), which is active in T-cells but not in B-cells. Instead, we focus on Igll1 and Myc, which are repressed by IKAROS with fast kinetics, and by a mechanism that primarily depends on CHD4, and secondarily on HDAC activity. Importantly, not all IKAROS target genes are repressed by IKAROS. Our manuscript illustrates this by showing increasing chromatin accessibility, RNAP2 occupancy and transcriptional output for the IKAROS target gene Zfp36 in Figure 2, and D). We have placed more emphasis on this in the revised manuscript by including the sentence “the molecular determinants of whether IKAROS binding results in gene activation or repression remain to be explored”. We also point out that reduced RNAP2 occupancy, reduced accessibility and transcriptional repression “were selective for Ikaros-repressed genes, as illustrated by comparison with the Ikaros-induced gene Zfp36 (Figure 2)”.

ii) The link between IKAROS-mediated gene activation and P-TEFb (Bottardi et al., Nucleic Acids Res. 2011, 39: 3505–3519) is very interesting. We have not explored the biochemical interaction between IKAROS and P-TEVb directly. Among IKAROS-repressed genes, few are regulated at the level of RNAP2 elongation (Table 1). However, in line with your comment and with the findings of Bottardi et al., we saw a significant increase in RNAP2 occupancy at 215 genes that were upregulated by IKAROS (Table 1). Of these, 202 showed an increase of RNAP2 at both the TSS and the gene body, indicating that increased recruitment of RNAP2 to the TSS was accompanied by a corresponding increase in elongation events. Only 13 upregulated genes with increased RNAP2 binding showed an accumulation of RNAP2 at the TSS (Table 1). These data are consistent with a role for IKAROS in promoting transcriptional elongation at upregulated genes that show increased RNAP2 occupancy.

iii) We have analysed the potential impact of the Polycomb repressive complex PRC2 (Oravecz et al., 2015) and found no increase in the PRC2-catalysed H3K27me3 mark at IKAROS-repressed genes within the 24 hour timeframe analysed in our study. This fact is indicated by the following sentence in our manuscript: “We saw no detectable increase in the trimethylation of H3K27 by PRC2 (Oravecz et al., 2015) at the Igll1 and Myc promoters 24h after nuclear translocation of Ikaros”.

iv) Our previous data provide support for the idea that both IKAROS and EBF1 bind the Igll1 promoter in large pre-B cells (Ferreiros-Vidal et al., 2013; Thompson et al., 2007). As pointed out by the referees, binding of Ikaros to the Igll1 promoter in large pre-B cells could potentially indicate a positive effect of IKAROS on Igll1 in large pre-B cells. To test this idea, we expressed a dominant negative form of Ikaros (IKAROS 159A). If IKAROS activates Igll1 in large pre-B cells, IKAROS 159A would be predicted to reduce Igll1 expression in large pre-B cells. Conversely, if IKAROS represses Igll1 in large pre-B cells, IKAROS 159A would be predicted to increase Igll1 expression. IKAROS 159A increased the expression of Igll1 (Thompson et al., 2007), demonstrating that Ikaros does not activate Igll1 in large pre-B cells.

v) To address this point, we have changed our Results section to explicitly state that reduced RNAP2 occupancy, reduced accessibility and transcriptional repression “were selective for Ikaros-repressed genes, as illustrated by comparison with the Ikaros-induced gene Zfp36 (Figure 2)” and modified our discussion to a) acknowledge that “the molecular determinants of whether IKAROS binding results in gene activation or repression with fast or slow kinetics remain to be explored” and b) provide a more explicit account of our mechanistic interpretation of the data: “Mechanistically, Ikaros can reduce EBF1 binding to the Igll1 promoter only in the presence of CHD4, and EBF1 is binding is sensitive to local chromatin structure (Treiber et al., 2010)”. “For genes that are expressed by Ikaros with fast kinetics, we favour a model where transcriptional repression by IKAROS involves active, CHD4- mediated chromatin remodeling of the Igll1 and Myc target gene promoters. This restricts promoter access, excludes RNAP2 and the transcriptional activator EBF1, and ultimately limits binding of Ikaros itself”.

3) Presentation and use of genome-wide data. The reviewers found that the integration of genome-wide information was potentially useful but incomplete; the quality of genome-wide information should be indicated with standard statistical measures (e.g. how much agreement is found among the triplicate MNase experiments? reproducibility of peaks etc.), even if the analysis of the entire dataset is being prepared in a separate study. MNase digestion patterns at two loci discussed, but overall significance is difficult to assess; is variation indicative of a change in nucleosome position, occupancy or other aspect of the individual experiments? It is not clear if the RNA pol II genome-wide study was similarly performed in triplicate. A related point to consideration of genome-wide data is related to point 2; it is reported that out of the 924 genes repressed upon increase in Ikaros expression, 372 show a reduction in RNAP2. How are the other 550 genes regulated? Are these not directly repressed by the Ikaros-NuRD complex? It would help this paper to provide enough information to understand Table 1 and Supplementary Data Table 1, which enumerate and list genes affected by Ikaros. For instance, the two genes studied in detail here are direct targets of Ikaros, and show a loss of RNA Pol II. A large number, but not the majority of other repressed genes also show a loss of Pol II; which of these are expected to be direct targets? An additional point regarding the table is the lack of clarity of how genes are assigned to it – is for example a log2 value of 0.05 even a significant change?

Author response Table 1 shows r2 correlation values between RNAP2 ChIP-seq replicates and experimental groups (4 independent biological replicates for control cells and 4 independent biological replicates for IKAROS-induced cells) as a standard statistical measure of reproducibility.

Genes*TSS**Gene bodies**Mean of replicates0.9900.9650.977Mean of exp. groups0.9700.9140.931

Author response Table 1. Reproducibility of RNAP2 ChIP-seq. Correlation values between RNAP2 ChIP- seq replicates and experimental groups, 4 independent biological replicates for control cells and 4 independent biological replicates for IKAROS-induced cells. * Correlation values between RNAP2 ChIP-seq data for the 500 genes with the greatest total variation in RNAP2 ChIP-seq signal

across all samples, including replicates and experimental groups. ** Correlation values between RNAP2 ChIP-seq at all transcription start sites (TSS) and gene bodies that showed statistically significant differences in RNAP2 ChIP-seq signal between experimental groups.

Figure 8 shows correlation values between MNase-seq replicates and experimental groups (3 independent biological replicates for control cells and 3 independent biological replicates for IKAROS-induced cells) at the indicated loci (left) and a graphical representation of base-pair level correlations for MNase sensitivity at the Igll1 promoter.

Author response image 1.Reproducibility of MNase-seq.Correlation values between MNase-seq replicates and experimental groups at the indicated loci, 3 independent biological replicates for control cells and 3 independent biological replicates for IKAROS-induced cells (left). Base-pair level correlation for MNase sensitivity at the Igll1 promoter (right).**DOI:**
http://dx.doi.org/10.7554/eLife.22767.025

We have added Figure 2—figure supplement 1 to the revised manuscript to demonstrate the reproducibility of RNAP2 ChIP-seq and MNase-seq replicates.

Our study reports on 4 independent biological replicates of RNAP2 ChIP-seq data. We apologise for not making this clear in the original manuscript, and now indicate this clearly in the revised figure legends.

The focus of our analysis is on downregulated genes that show a significant reduction in RNAP2 binding, and the great majority of these show no evidence for increased RNAP2 pausing. The referees point out that of 924 genes downregulated at adj. p<0.05, only 372 (40.3%) showed a significant reduction in RNAP2 ChIP-seq signal. For most of the remaining downregulated genes, the reduction in RNAP2 binding did not reach statistical significance. To test whether this is due to a difference in sensitivity between RNA-seq and RNAP2 ChIP-seq, we repeated our analysis considering only genes that were significantly downregulated and also showed a minimal log2 fold- change of 2. Of 174 genes that met these criteria, 95 showed a significant reduction in RNAP2 ChIP- seq signal (55%). The percentage of downregulated genes with significantly reduced RNAP2 ChIP- seq signal further increased to 62% when considering only genes downregulated with adj. p<0.01. The finding that the majority of strongly downregulated genes had reduced RNAP2 ChIP-seq signals supports the idea that downregulated expression without reduced RNAP2 ChIP-seq signals is due at least in part to a difference in sensitivity between RNA-seq and RNAP2 ChIP-seq. Regardless of the criteria applied to define downregulated genes, reduced RNAP2 elongation remained an exception among IKAROS/NuRD-repressed genes: 4% (15/372) at adj. p<0.05, 5.4% (5/93) at log2FC>2 and p<0.05, and 4.6% (4/87) at log2FC>2 and p<0.01.

Differential gene expression was defined by standard statistical criteria, namely a p-value of <0.05 after Benjamini-Hochberg correction for multiple testing (adj. p<0.05). We have added this information to the revised legend of Table 1 and Table 2 and to the methods section.

4) Modeling. The reviewers had differing opinions about the modeling; some asked for further validation through additional measurements of actual nucleosome positioning and Ikaros occupancy at individual sites, while another view was that the models are useful enough to point toward future studies of mechanism. A justification of the application of the modeling would strengthen this aspect of the study. In addition, a clarification of the actual mechanism suggested is requested: Ikaros has been variously described as a recruiter of NuRD, or an antagonist, or binding in an overlapping manner at promoters but not enhancers. The kinetics of binding shown in 4D indicate that a model with simple, stable recruitment of NuRD by Ikaros is too simple. But it is not clear what mechanism the authors are proposing, based on their findings.

The experimental finding that Ikaros binding increases prior to the displacement of EBF1 is based on 9 separate chromatin preparations and 9 separate ChIP experiments for each factor (3 shown in Figure 1, Figure 3 controls and 3 sh*Chd4* experiments shown in Figure 7). In modeling these data, we cannot find parameters that would allow additional binding of Ikaros without simultaneous displacement of EBF1 in a 2-state model (Figure 1). We are therefore confident that we can reject 2 state models.

Three-state models such as the one shown in Figure 1 can explain increased Ikaros binding prior to the displacement of EBF1. The parameters used are not critical, as long as the binding of Ikaros is as strong as or stronger than the binding of EBF1, which his seems reasonable given that Ikaros binds as an obligate dimer and a potential multimer.

The 3-state model in Figure 1 and Figure 1—figure supplement 1 differ with respect to the increase in nuclear Ikaros. In Figure 1 Ikaros increases in a single step. In Figure 1—figure supplement 1 Ikaros increases in several steps. This is based on immunofluorescence staining with antibodies to the Ikaros HA tag and semi-quantitative measurements of confocal images such as the ones shown in Figure 1. The prediction that Ikaros binding should rise as a function of increasing nuclear concentration of Ikaros is robust to changes in parameters.

The 4-state model in Figure 1—figure supplement 1 assumes that nucleosome binding is more stable than TF binding. The prediction made by Figure 1—figure supplement 1/B is that loss of nucleosome remodeling will result to increased binding of Ikaros and reduced displacement of EBF1. To test this prediction, we performed ChIP experiments in control and sh*Chd4* cells. As shown in Figure 7, Ikaros binding was increased, and the displacement of EBF1 was reduced after Chd4 knockdown. Hence, our conclusion that chromatin remodeling limits the binding of Ikaros and EBF1 is based on experimental data, and the models in Figure 1—figure supplement 1/B illustrate how this finding might be explained.

We have added a new section that details the equations and parameters used for the modeling to the methods section of the revised manuscript. As requested, we have also added a paragraph summarising the mechanism we propose to explain our findings to the Discussion section of the revised manuscript: “we favour a model where transcriptional repression by IKAROS involves active, CHD4- mediated chromatin remodeling of the Igll1 and Myc target gene promoters. This restricts promoter access, excludes RNAP2 and the transcriptional activator EBF1, and ultimately limits binding of Ikaros itself”.

5) Endogenous Ikaros vs. induced form. An increase in Ikaros nuclear localization is assumed from the immunofluorescence data but this is not a very quantitative method. A western blot analysis for Ikaros in the nuclear vs, cytoplasmic fractions at the different time points should be performed. The way the ChIP data is presented, it is not clear how much endogenous Ikaros is already present at the target promoters before the induction of the tamoxifen-regulated protein, and how this compares with the induced levels. Previous studies have already noted that Ikaros is present at these genes, so how much of the modifications are due to overexpression of the protein? Why would nucleosome changes results only from overexpression; is this a function of rapid interconversion between active and inactive states on the genes with only endogenous Ikaros? In addition, the paper is silent about the Ikaros-like Aiolos protein; is it present at these genes, and does it play a role in regulation under these conditions?

As requested, Figure 9 shows endogenous versus exogenous Ikaros expression by western blotting for Ikaros in whole cell lysate, cytoplasmic and nuclear fractions before and after 4- OHT addition.

Author response image 2.Endogenous versus exogenous Ikaros expression.Western blotting for exogenous Ikaros (anti-HA), total Ikaros (Ikaros C-terminal antibody), the cytoplasmic marker Tubulin and the nuclear marker histone H3 in whole cell lysate (W), cytoplasmic (C) and nuclear (N) fractions before and after 4-OHT addition. Mobility distinguishes endogenous Ikaros from the HA-Ikaros-ERt2 fusion protein.**DOI:**
http://dx.doi.org/10.7554/eLife.22767.026

We have added the occupancy by endogenous Ikaros to the excel spread sheets listing the primary data for Figure 1 and Figure 7. Figure 7 uses IKAROS_ERt2-IRES-Cherry, and 4-OHT increases Ikaros binding to Igll1 by 170% over endogenous Ikaros at 5 minutes and by 240% at 15 minutes. Figure 1 uses the more highly expressed IKAROS_ERt2-IRES-GFP, and 4-OHT increases Ikaros binding to Igll1 by 390% over endogenous Ikaros at 5 minutes and by 540% at 15 minutes. Both IKAROS-IRES-GFP and IKAROS-IRES-Cherry result in increased nucleosome occupancy, reduced RNAP2 occupancy and reduced transcriptional output, indicating that a moderate increase in Ikaros binding is sufficient to trigger the described responses.

Ikaros dosage in pre-B cells drives changes in gene expression that resemble the in vivo differentiation from large (Fr.C') to small (Fr.D) pre-B cells (Ferreiros-Vidal et al., 2013). Ikaros is essential for the differentiation of large (Fr.C') to small (Fr.D) pre-B cells (Heizmann et al., 2013; Joshi et al., 2014; Schwickert et al., 2014), and the majority of Ikaros target genes are already bound by Ikaros in large (Fr.C') pre-B cells (Ferreiros-Vidal et al., 2013). The Ikaros family member Ikzf3 is upregulated between the large (Fr.C') and small (Fr.D) pre-B cell stage (Thompson et al., 2007), and increased Ikaros/Aiolos expression drives increased binding to target gene promoters (Thompson et al., 2007; Ferreiros-Vidal et al., 2013). Hence, Ikaros dosage is critical for the impact of Ikaros on gene expression. Inducible translocation of Ikaros into the nucleus is intended to model the increase in the expression of Ikaros family proteins during the differentiation of large (Fr.C') to small (Fr.D) pre- B cells. Accordingly, our system aims to moderately increase Nuclear Ikaros dosage, as illustrated by western blotting for Ikaros in whole cell lysate, nuclear and cytoplasmic fractions before and after 4- OHT addition (Figure 9).

These data indicate that nuclear Ikaros dosage translates into the regulation of Ikaros target genes, including Igll1, Myc, and other genes where Ikaros binding is seen in large pre-B (Fr.C') cells.

Nucleosome changes and repression of Igll1 and Myc result from an increase in the nuclear dosage and increased target gene binding of Ikaros and/or Aiolos. In B cell differentiation in vivo, both Igll1 and Myc are downregulated as Fr.C' cells progress to Fr.D (www.immgen.org).

Both (highly expressed) IKAROS-IRES-GFP and (moderately expressed) IKAROS-IRES-Cherry result in increased nucleosome occupancy, reduced RNAP2 occupancy and reduced transcriptional output. This indicates that a moderate increase in Ikaros expression and binding is sufficient to trigger the described responses.

To explore the relationship between endogenous and exogenous Ikaros we repeated Ikaros induction experiment in B3 cells in which we targeted both endogenous Ikzf1 alleles by CRISPR/Cas9. In these cells, repression of Igll1 and Myc (and induction Zfp36) occurred after Ikaros-ERt2 induction with kinetics similar to those we found in B3 cells with intact endogenous Ikzf1 alleles (Figure 10). These data indicate that Ikaros-mediated transcriptional repression can occur with fast kinetics in the absence of endogenous Ikaros.

Author response image 3.Transcriptional regulation by Ikaros-ERt2 in the absence of endogenous Ikaros.B3 cells with disrupted endogenous Ikzf1 alleles were transduced with Ikaors-ERt2. Primary transcripts for Igll1, Myc, and Zfp36 were measured at the indicated times after 4-OHT addition. Mean**DOI:**
http://dx.doi.org/10.7554/eLife.22767.027

± SE of 3 independent biological replicates.

</Figure 10 title/legend>

BLNK-mediated upregulation of Ikzf3 results in the repression of Igll1 (Thompson et al., 2007), and Ikaros and Aiolos both repress Myc (Ma et al., 2010).

6) MNase interpretation. The purported identification of "fragile" sites by MNase tests was not viewed as very strong, for a number of reasons. To identify such changes as an alternatively-bound form of the nucleosome, one would wish for a titration of MNase, not just a single digestion. In addition, it would be important to know if the change is due to a nucleosome shifting in position, rather than being weakly bound. Finally, the differential amplification in 6B with "short" and "long" amplicons may or may not be significant – it is not clear if the n=3 is a technical replicate, in which case, the support is weak.

As discussed in section 1, our conclusions are based on independent biological replicates and the data are solid. MNase titration followed by quantitative PCR confirmed that TSA treatment reduced the protection of long amplicons at the Igll1 promoter 24h after Ikaros induction (Figure 11). Nevertheless, without additional sequencing experiments we cannot be certain whether histone deacetylase inhibition reduces nucleosome stability or alters nucleosome positioning. We have therefore modified the narrative of our revised manuscript as follows: “These data suggest that in the absence of HDAC activity, nucleosomes at the Igll1 promoter and TSS may remain unstable (Henikoff et al., 2011) or less evenly spaced”. What is clear, however, is that (i) Ikaros changes nucleosome occupancy of the Igll1 promoter and (ii) HDAC inhibition affects nucleosome occupancy of the Igll1 promoter after 24 hours.

Author response image 4.Titration experiments confirm the impact of HDAC inhibition on MNase protection of *Igll1* promoter chromatin.MNase titration followed by quantitative PCR showed that 1ng/ml TSA for 24 hours did not significantly affect protection of 80-120 bp amplicons (short, left) but significantly reduced protection of 130-140 bp amplicons (long, right) at the Igll1 promoter. Mean ± SE of 3 independent biological replicates.**DOI:**
http://dx.doi.org/10.7554/eLife.22767.028

A number of technical points relating to reagents, methods, and data presentation should be addressed:

7) The antibodies for Ikaros and EBF1 are not described; in the Materials and methods, a reference is made to Ferreiros-Vidal 2013, but EBF1 is not used there. What is the measure of specificity for these reagents? Is there other data to support the EBF1 binding shown here?

We apologise for these omissions and have added the following sentences to the methods section: “Ikaros ChIP was as described (Ferreiros-Vidal et al., 2013) using a C-terminal Ikaros antibody provided by Dr Stephen Smale that has been extensively characterised in ChIP-seq experiments (Ferreiros-Vidal et al., 2013; Bossen et al., 2015). We used an EBF1 antibody (Santa Cruz sc- 137065) previously characterised by ENCODE for EBF1 ChIP-seq”. Additional support for the Ikaros and EBF1 binding shown here comes from Schwickert et al., 2014, for EBF1 (using a V5 epitope tag- specific reagent), Ferreiros-Vidal et al., 2013, for HA-Ikaros (using a HA epitope tag-specific reagent), Ferreiros-Vidal et al., 2013, for endogenous Ikaros (using the C-terminal Ikaros antibody employed in this study) and Bossen et al., 2015, for endogenous Ikaros (also using the C-terminal Ikaros antibody employed in this study) as summarised in Figure 12.

Author response image 5.Additional support for Ikaros and EBF1 binding at the *Igll1* promoter in B cell progenitors.V5 epitope tag-specific EBF1 ChIP and input (top, Schwickert et al., 2014), ChIP for endogenous Ikaros using the C-terminal Ikaros antibody employed in this study (Bossen et al., 2015; Ferreiros-Vidal et al., 2013), HA-tagged Ikaros using a HA epitope tag-specific reagent (Ferreiros-Vidal et al., 2013) and input (Ferreiros-Vidal et al., 2013) show binding of EBF1 and Ikaros to the *Igll1* promoter.**DOI:**
http://dx.doi.org/10.7554/eLife.22767.029

8) In Figure 1, "enrichment" is the measure of chromatin binding for Ikaros and EBF1. The Materials and methods don't describe how these calculations were performed. One way might be to measure the signal at some non-bound site, and divide the ChIP signal% input by this background control. If so, what is their defined background? What were the% input values for these promoters and other locations?

Figure 1 and Figure 7 compare ChIP signals (% input) at each time point to ChIP signals (% input) before 4-OHT addition (0min). We measured ChIP signals at control loci as part of our quality control, but not for the purpose of normalisation. EBF1 and Ikaros ChIP signals for the pluripotency-related Rex1 locus are shown in Figure 13.

Author response image 6.EBF1 and Ikaros ChIP at *Igll1* compared to the control locus *Rex1*.The data shown are for B cells transduced with Ikaros-IRES-Cherry and represent the mean ± SE of 3 independent biological replicates. The data have been added to the source data file for Figure 7 of the revised manuscript.**DOI:**
http://dx.doi.org/10.7554/eLife.22767.030

</Figure 13 title/legend>

9) For Pol II and TFIIB measurements, the manuscript shows directly the% input recovered from ChIP, which is a preferable approach for chromatin aficionados (their values ~1% are quite credible). The Y-axis is labeled "enrichment", however, which is confusing, because presumably the% input was not normalized to signal at another locus.

All ChIP-PCR data in our manuscript are% input or fold-changes thereof. “Enrichment” was added to the Y-axes simply to indicate that these are ChIP experiments, not to indicate normalisation to a reference locus. We have changed the relevant axes to “% input”.

10) In Figure 3—figure supplement 1, part B, using the TEV-activated Ikaros system, depletion of Pol II does further affect MNase sensitivity, which is different from the system with 4-OHT induced Ikaros. The authors don't comment on this, though they previously make the point that Pol II eviction is not causing the changes in MNase sensitivity.

The main points of this experiment are to show (i) that Igll1 promoter accessibility is maintained in the absence of RNAP2 and (ii) Ikaros induction can still induce chromatin remodeling in triptolide-treated cells. Figure 3—figure supplement 1, part B supports these conclusions. Thank you for pointing out that Ikaros induction via split TEV induction causes an even greater increase in nucleosome occupancy of the Igll1 promoter, suggesting synergy between Ikaros induction and RNAP2 depletion. We have added the following sentence to the legend of Figure 3—figure supplement 1, part B: “Ikaros induction and tripolide- mediated RNAP2 depletion may synergise in increasing the nucleosome occupancy of the Igll1 promoter”.

11) Figure 5—figure supplemental Figure part A: TSA treatment apparently induces a decrease in histone acetylation levels? Please comment.

As the referees point out, TSA treatment can have moderate effects on local histone acetylation prior to Ikaros induction (Figure 5—figure supplement 1), as well as local RNAP2 occupancy and primary transcript levels prior to Ikaros induction (Figure 5). This is consistent with proposals that acetylation/deacetylation cycles may promote transcription (Wang et al., Cell 2009, 138: 1019-1031; Perissi et al., Nat Rev Genet 2010, 11: 109–123). Importantly, TSA increased global H3 and H4 acetylation (Figure 5—figure supplement 1, left) and ChIP experiments in the same panel show that TSA prevents Ikaros-induced local deacetylation of H3 and H4 at the Igll1 and Myc promoters. In the cells used for these ChIP experiments, TSA did not interfere with the Ikaros-induced increase in nucleosome occupancy, reduction in RNAP2 occupancy, and transcriptional repression (Figure 5), supporting our conclusion that HDAC activity is not required for the Ikaros-induced increase in nucleosome occupancy, reduction in RNAP2 occupancy, and transcriptional repression.

12) The differences for pericentric chromatin localization +/- TSA (6D) and loading of Ikaros with or without CHD4 (7B) may or may not be significant; it is not clear how reproducible these differences are in independent experiments. (See Point 1). In 6D, apparently two experiments were conducted; is this data from one?

We have performed additional DNA-FISH experiments, and the revised Figure 6 and Figure 7 now show results for 3 independent biological replicates with mean and SE. A total of 1233 alleles were scored, 355 for control cells, 314 after Ikaros induction, 248 for Ikaros in the presence of TSA, and 316 for Ikaros in sh*Chd4* cells. The p-values as calculated based on replicates using GLM (binomial logit) were 9.54x10-18 for the effect of TSA and 5.54x10-38 for the effect of Chd4 knockdown, indicating that the effect of TSA and Chd4 knockdown on the positioning of Igll1 alleles were statistically significant. The figures, legends and source files of the revised manuscript have been updated accordingly.

13) It is not clear why the KinTec modeling used the specific parameters indicated in Materials and methods. Have these been measured in a particular system?

As discussed in detail in section 4, the models were largely robust to parameter choices, and our choices were based on the greater stability of nucleosomes compared to transcription factors, and the knowledge that Ikaros binds as an obligate dimer/facultative multimer. The rate of nuclear Ikaros was estimated based on semi-quantitative confocal microscopy.

14) In subsection “Loss of RNAP2, reduced promoter accessibility, and transcriptional repression are early and near-simultaneous events” the authors state that they use primer pairs that span introns as a means to measure unspliced transcripts. They presumably mean primers that span individual exon/intron junctions? The primers used should be included in Materials and methods.

Thank you for pointing out the inaccurate phrasing. The primer pairs used were “designed to span an intron-exon boundary”. We have corrected this in the revised manuscript. Primer sequences are listed in the Methods section of the revised manuscript.

[Editors' note: further revisions were requested prior to acceptance, as described below.]

The reviewers had asked for more specific analysis of genome-wide gene expression, since only a minority of genes showed the Pol II decrease found at the two promoters of interest. In the response to the reviewers, the authors reanalyze the data, and conclude that for a majority of promoters, it appears that Pol II decrease at the promoter is the trend, and that differences in sensitivity of measuring mRNA vs. Pol II occupancy may explain some of the discrepancy.

1) In addition to having this information in the letter of response, the manuscript should indicate that they believe the majority of genes are showing reductions in Pol II, but that differential sensitivity may be an issue. Otherwise it seems the authors are content to ignore what might be a mechanism that impacts the majority of repressed genes.

We have changed the narrative to “Genome-wide, RNAP2 occupancy was significantly reduced at 372 downregulated genes 6 hours after Ikaros translocation (Table 1)” and added to following information to the legend of Table 1: “RNAP2 occupancy was significantly decreased at only 40.3% (372 of 924) of genes downregulated at adj. p<0.05 but increased to 62% when considering only genes downregulated with adj. p<0.01 and a minimal log2 fold-change of 2. This suggests that the failure to detect decreased RNAP2 occupancy at the majority of downregulated genes may be due to the limited sensitivity of RNAP2 ChIP-seq compared to RNA-seq”.

2) The Western blot data should be included in the manuscript as part of, or attached to, Figure 1, where the system is introduced.

We have added a western analysis showing the levels of native and induced Ikaros over time to Figure 1—figure supplement 1. A description of the experiment has been added to the figure legend, and the figure is referred to in the main text where the system is introduced.

3) The text still refers to "intron-spanning primers" – that needs to be fixed.

We have fixed the offending sentence on p10 of the manuscript to read: “We used quantitative RT-PCR with primers spanning intron-exon boundaries to quantify the abundance of unspliced (primary) transcripts…”

4) Typo Figure 7 Average nuceosome profile

We have corrected the typo in Figure 7 to “nucleosome profile”